# RASSF1A is required for the maintenance of nuclear actin levels

Maria Chatzifrangkeskou[1] (ID), Dafni-Eleftheria Pefani[1,2,3] (ID), Michael Eyres[1], Iolanda Vendrell[1,4], Roman Fischer[4], Daniela Pankova[1] & Eric O'Neill[1,*] (ID)

## Abstract

Nuclear actin participates in many essential cellular processes including gene transcription, chromatin remodelling and mRNA processing. Actin shuttles into and out the nucleus through the action of dedicated transport receptors importin-9 and exportin-6, but how this transport is regulated remains unclear. Here, we show that RASSF1A is a novel regulator of actin nucleocytoplasmic trafficking and is required for the active maintenance of nuclear actin levels through supporting binding of exportin-6 (XPO6) to RAN GTPase. *RASSF1A* (Ras association domain family 1 isoform A) is a tumour suppressor gene frequently silenced by promoter hypermethylation in all major solid cancers. Specifically, we demonstrate that endogenous RASSF1A localises to the nuclear envelope (NE) and is required for nucleocytoplasmic actin transport and the concomitant regulation of myocardin-related transcription factor A (MRTF-A), a co-activator of the transcription factor serum response factor (SRF). The RASSF1A/RAN/XPO6/ nuclear actin pathway is aberrant in cancer cells where *RASSF1A* expression is lost and correlates with reduced MRTF-A/SRF activity leading to cell adhesion defects. Taken together, we have identified a previously unknown mechanism by which the nuclear actin pool is regulated and uncovered a previously unknown link of RASSF1A and MRTF-A/SRF in tumour suppression.

**Keywords** exportin-6; MRTF-A; nuclear actin; nuclear envelope; RASSF1A
**Subject Categories** Cell Adhesion, Polarity & Cytoskeleton; Membrane & Intracellular Transport; Signal Transduction
**The EMBO Journal (2019) 38: e101168**

## Introduction

Actin is one of the most highly conserved cytoskeletal proteins and found in all eukaryotic cells. The fundamental roles of actin are critical for biological processes such as determination of cell shape, vesicle trafficking and cell migration processes that are often deregulated in transformed cells. Whilst the role of cytoplasmic actin is well established, the presence of nuclear actin is increasingly appreciated to play a crucial role in cellular responses to internal and external mechanical force (Guilluy *et al*, 2014). Nuclear actin is actively imported and exported through the activity of importin-9 (Dopie *et al*, 2012) and exportin-6 (Stuven *et al*, 2003) and has been implicated in transcription (Egly *et al*, 1984; Hofmann *et al*, 2004), cellular differentiation (Sen *et al*, 2015) and DNA repair (Belin *et al*, 2015). Nuclear actin monomers have been shown to directly regulate the myocardin-related transcription factor A (MRTF-A), a mechanosensitive co-factor of the serum response factor (SRF) transcription pathway (Vartiainen *et al*, 2007; Baarlink *et al*, 2013).

The Ras association domain family (RASSF) genes are upstream regulators of the Hippo tumour suppressor pathway. The family is composed of 10 members which are divided into two subgroups: (i) RASSF 1–6 possess a Ras association (RA) domain and a SARAH (Sav/Rassf/Hpo) protein–protein interaction domain in the C-terminus, and (ii) RASSF 7–10 also contain a RA domain at the N-terminus but lack a recognisable SARAH domain (Sherwood *et al*, 2010). The RA domains of the RASSF family bind K-RAS, H-RAS, RAP1/2 and RAN GTPases with varying affinity (Avruch *et al*, 2006; Dallol *et al*, 2009). The RASSF1A isoform is a *bona fide* tumour suppressor gene whose inactivation is implicated in the development of a wide range of human tumours including breast, lung and gastrointestinal cancer (Grawenda & O'Neill, 2015). Although gene deletion and germline mutations exist, the most widespread loss of RASSF1A function occurs through promoter hypermethylation-associated transcriptional silencing (Grawenda & O'Neill, 2015). RASSF1A directly binds Hippo kinases, mammalian sterile 20-like kinases 1 and 2 (MST1 and MST2) through the SARAH domain, promoting downstream Hippo pathway signalling to YAP1 (Guo *et al*, 2007; Matallanas *et al*, 2007). In response to DNA damage, nuclear RASSF1A is phosphorylated on Ser131 by ATM (ataxia-telangiectasia-mutated) or ATR (ATM- and Rad3-related) kinases, which promotes dimerisation and trans-autophosphorylation of MST2 required for its activation (Hamilton *et al*, 2009; Pefani *et al*, 2014). Plasma membrane localisation of cytoplasmic RASSF1A occurs in response to growth factor signalling or direct RAS

1  Department of Oncology, University of Oxford, Oxford, UK
2  Laboratory of Biology, Medical School, National and Kapodistrian University of Athens, Athens, Greece
3  Biomedical Research Foundation of the Academy of Athens, Athens, Greece
4  Nuffield Department of Medicine, Target Discovery Institute, University of Oxford, Oxford, UK
  *Corresponding author. Tel: +44 1865617321; E-mail: eric.oneill@oncology.ox.ac.uk

association with the RA domain (Pefani *et al*, 2016). However, the localisation of RASSF1A to mitotic structures has been attributed to RA domain binding to tubulin (also a GTP-binding protein) and is a likely cause of hyperstabilisation of tubulin commonly observed in interphase cells overexpressing exogenous levels (Liu *et al*, 2003; Dallol *et al*, 2004; Vos *et al*, 2004; El-Kalla *et al*, 2010).

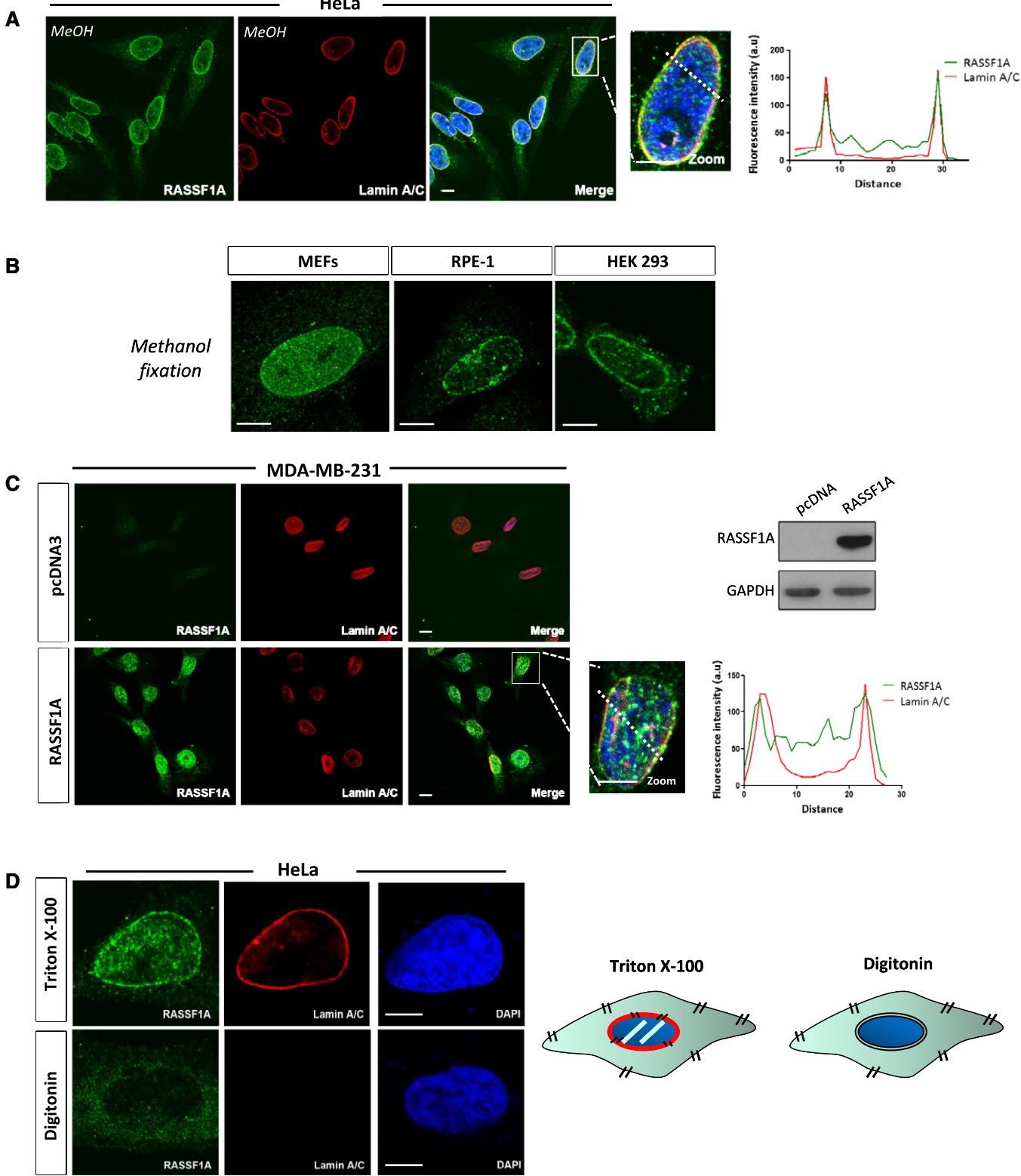

Figure 1.

**Figure 1.   RASSF1A localises at the inner nuclear membrane.**

A   Representative confocal images of RASSF1A localisation in HeLa cells. Nuclear DNA was stained with DAPI. Fluorescence intensity profile of Lamin A/C (red) and RASSF1A (green) signals across the HeLa nuclei. Position of line scan indicated by the dashed white line. Scale bars = 10 μm.

B   Immunofluorescence images of endogenous RASSF1A expression in mouse embryonic fibroblasts (MEFs) (left), human retinal pigmented epithelial RPE-1 cells (middle) and human embryonic kidney HEK293 cells (right). Scale bars = 10 μm.

C   Co-staining of endogenous *RASSF1A* expression in MDA-MB-231 cells transfected with the pcDNA3 expressing RASSF1A. Western blot analysis shows the protein levels of RASSF1A following plasmid transfection. Fluorescence intensity profile of Lamin A/C (red) and RASSF1A (green) signals across the HeLa nuclei. Position of line scan indicated by the dashed white line. Scale bars = 10 μm.

D   *Left:* representative confocal images of HeLa cells permeabilised either with Triton X-100 (top panels) or with digitonin (bottom panels) and stained for RASSF1A, Lamin A/C and DNA (DAPI). Scale bars = 10 μm. *Right:* schematic representation of the cellular membranes permeabilised with Triton X-100 and digitonin.

Source data are available online for this figure.

Here, we show that nuclear RASSF1A is recruited to the NE by a lamina-associated pool of the Hippo kinase, MST2. Furthermore, we find that RASSF1A is required for the association of exportin-6 (XPO6) with RAN GTPase, the nuclear export of the XPO6 cargoes, actin and profilin, an actin-binding protein involved in the dynamic sensing of the actin cytoskeleton and the regulation of the MRTF-A/SRF axis. Tumour cells lacking RASSF1A expression display high levels of nuclear actin/profilin, reduced MRTF-A levels and low transcription of SRF target genes, including SRF itself. In line with these findings, we find that human tumours display a high level of correlation between RASSF1A and SRF expression, with evidence for low SRF mRNA as a poor prognostic factor in breast and liver cancers. These events reveal the potential primary mechanism for the tumour suppressor RASSF1A in cancer being mediated through deregulation of nuclear actin transport.

## Results

### RASSF1A localises at the inner nuclear membrane

RASSF1A acts as a microtubule-associated protein that stabilises microtubules (Liu *et al*, 2003; Dallol *et al*, 2004; Vos *et al*, 2004; El-Kalla *et al*, 2010). However, these phenotypes were determined after the accumulation of overexpressed protein over a 24- to 48-h period. To identify the localisation of endogenous nuclear RASSF1A by indirect immunofluorescence (IF) and methanol fixation, we used an anti-RASSF1A antibody (ATLAS), the specificity of which was validated in cells depleted for RASSF1A (Figs EV1A and B). Interestingly, in HeLa cells endogenous RASSF1A is distributed predominantly throughout the nuclear interior and at the perinuclear regions. Co-staining of RASSF1A and the inner nuclear membrane (INM) protein Lamin A/C showed a high degree of co-localisation and similar fluorescence intensity profiles (Fig 1A). Silencing of RASSF1A resulted in loss of the nuclear ring staining, indicating that the fluorescence signal is specific for RASSF1A (Fig EV1B). Localisation was conserved across multiple cell types including mouse embryonic fibroblasts (MEFs), human retinal pigmented epithelial RPE-1 cells and human embryonic kidney HEK 293 cells (Fig 1B). The *RASSF1A* gene promoter is highly methylated in the breast carcinoma cell line MDA-MB-231 and therefore significantly downregulated (Montenegro *et al*, 2012). Concomitantly, no immunofluorescence was observed in MDA-MB-231 cells, but upon transient transfection with a plasmid expressing RASSF1A, NE localisation of the exogenous protein was evident (Fig 1C). Alternatively,

treatment of MDA-MB-231 with the demethylating agent 5′-aza-dC led to the re-expression of endogenous *RASSF1A* and yielded similar staining (Fig EV1C).

To further assess the localisation of RASSF1A in relation to the structure of the NE, HeLa cells were treated with either Triton X-100 or digitonin. Whereas Triton X-100 is used to permeabilise all cellular membranes, digitonin can permeabilise preferentially the plasma membrane of cultured cells and leave the NE intact. Therefore, only proteins of the outer nuclear membrane (ONM) that face the cytoplasm are detectable in digitonin-permeabilised cells. In Triton X-100-permeabilised HeLa cells, RASSF1A co-localises with Lamin A/C (Fig 1D). In contrast, neither Lamin A/C nor RASSF1A was detected at the nuclear envelope in digitonin-permeabilised cells. Permeabilisation of the plasma membrane of the digitonin-treated cells was validated using co-staining of RASSF1A with α-tubulin, a marker of cytoplasm (Fig EV1D). These results support the hypothesis that RASSF1A localises at the inner nuclear membrane (INM) of the NE.

The ATM and ATR kinases catalyse phosphorylation of RASSF1A at Ser131 to regulate its activity (Hamilton *et al*, 2009). Phosphorylated (Ser131) RASSF1A was present within the nucleus and at the INM of HeLa cells (Fig EV1E). To determine whether phosphorylation of RASSF1A on Ser131 by ATM and ATR kinases is required for its localisation at the inner nuclear membrane, HeLa cells were treated with the ATR inhibitor (VE-821) or ATM inhibitor (KU55933). Inhibition of ATR, ATM or combined ATR/ATM activity did not abolish RASSF1A perinuclear localisation (Fig EV1F). Collectively, the above observations show that the RASSF1A association with the NE is specific and not dependent on the kinase activity of ATR and ATM kinases.

### RASSF1A localisation at the NE is mediated by MST2

RASSF1A directly binds mammalian STE20-like protein kinases 1 and 2 (MST1 and MST2) to support maintenance of MST1/2 phosphorylation and activation (Sánchez-Sanz *et al*, 2016). Interestingly, mass spectrometry analysis of MST2 eluted fractions from HeLa lysates identified confidently the interaction between MST2 and Lamin A/C (Figs 2A and EV2A) and suggested a potential interaction with Lamin B1. We further examined the cellular localisation of nuclear MST1/2 kinases by IF in HeLa cells (with methanol fixation) and we observed that only MST2 (ab71960), and not MST1, exhibited perinuclear distribution similar to RASSF1A, as shown with co-staining with Lamin B1 (Fig 2B). The perinuclear staining of MST2 was confirmed using another antibody (ab52641) recognising the

**A**

| | | Biological replicate 1 | | Biological replicate 2 | | Biological replicate 3 | |
|---|---|---|---|---|---|---|---|
| Accession No | Name | Mascot score | No Significant peptides (unique) | Mascot score | No Significant peptides (unique) | Mascot score | No Significant peptides (unique) |
| Q13188 | Serine/threonine-protein kinase 3 | 397 | 18 (13) | 773 | 26 (21) | 555 | 19 (12) |
| Q13043 | Serine/threonine-protein kinase 4 | 284 | 9 (5) | 444 | 14 (7) | 334 | 12 (5) |
| Q9NS23 | Ras association domain-containing protein 1 | 86 | 4 (4) | 88 | 6 (6) | 92 | 9 (9) |
| P20700 | Lamin-B1 | - | - | 62 | 2 (1) | - | - |
| P02545 | Pre-lamin A/C | 328 | 12 (12) | 447 | 15 (15) | 251 | 7 (7) |

**B**

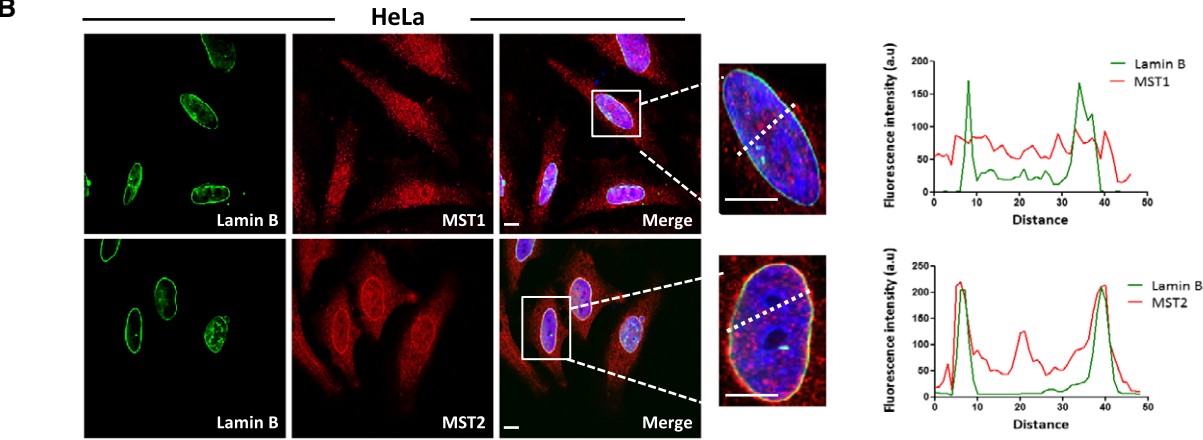

**C**

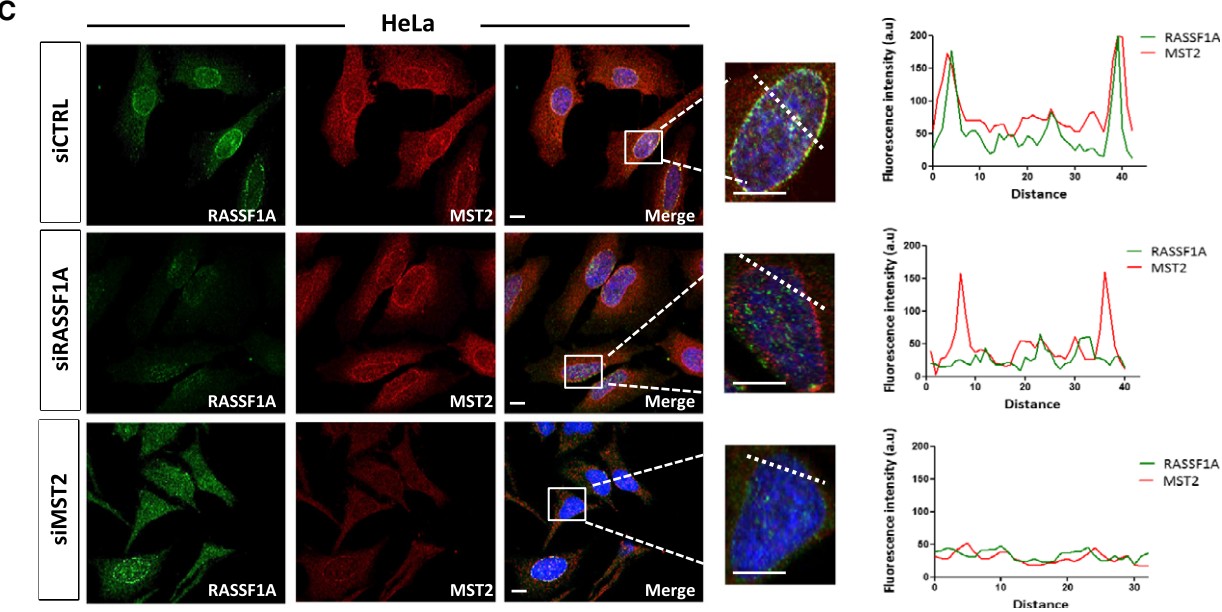

**Figure 2.**

◀

**Figure 2. RASSF1A localisation at the NE is mediated by MST2.**

A List of proteins identified by LC-MS/MS in MST2 IP. The table includes the MASCOT score and number of significant peptides and unique peptides (in brackets) identified in each biological replicates after applying a 20 ion cut-off and 1% FDR rate. Lamin B1 was identified only in 1 out of the 3 biological replicates with 1 unique peptide. However, interactions were verified by WB (Fig EV2A).

B Representative confocal images of MST1 (top) and MST2 (bottom) co-stained with the nuclear envelope marker Lamin B. DNA was stained with DAPI. Fluorescence intensity profile of Lamin B (green) and MST1 or MST2 (red) signals across the HeLa nuclei. Position of line scan indicated by the dashed white line. Scale bars = 10 µm.

C Immunofluorescence images of RASSF1A and MST2 in siCTRL-, siRASSF1A- and siMST2-treated cells. DNA was stained with DAPI. The graphs illustrate the fluorescence intensity profile of Lamin B (green) and MST2 (red) signals along the white lines shown in the merged panels. Scale bars = 10 µm.

N-terminus (Fig EV2B). To explore the role of MST2 localisation at the NE in relation to RASSF1A, we next reduced MST2 expression using siRNA-mediated knockdown and observed substantial reduction in RASSF1A staining at the INM in the absence of MST2 without affecting the overall nuclear RASSF1A protein levels (Fig EV2C and D). Conversely, RASSF1A knockdown did not affect MST2 NE distribution (Fig 2C). Overall, these results suggest that MST2 mediates RASSF1A localisation at the INM via lamina association.

## RASSF1A binds to XPO6 through its SARAH domain

We previously showed that RASSF1A interacts with RAN GTPase (Dallol *et al*, 2009), a key component in the regulation of nucleocytoplasmic transport. Given our data above showing the localisation of RASSF1A in the INM, we now sought to identify whether RASSF1A plays an active role in the process of RAN-dependent nuclear export. To test our hypothesis, we performed immunoprecipitation using extracts of HeLa cells and antibodies against exportin-1 (CRM1/XPO1), exportin-4 (XPO4), exportin-5 (XPO5), exportin-6 (XPO6) and exportin-7 (XPO7; Fig 3A). Interestingly, we found that RASSF1A specifically interacts with XPO6, but not with CRM1/XPO1 which is the most conserved and responsible for export of a wide variety of cargoes (Fornerod *et al*, 1997).

In order to determine the domain responsible for its interaction with XPO6, we exogenously expressed full-length MYC-RASSF1A and different RASSF1A-truncated mutants in HeLa cells. Whereas full-length MYC-RASSF1A (aa 1–340) and MYC-RASSF1A (aa 120–340) could specifically precipitate XPO6, no signal was detected with mutants lacking the SARAH domain, MYC-RASSF1A (aa 1–288) and MYC-RASSF1A (aa 120–288; Fig 3B). Overall, these data demonstrate that the SARAH domain of RASSF1A is required for the interaction with XPO6.

We further investigated the role of RASSF1A on the association of RAN with XPO6. Most strikingly, RASSF1A appeared to be required to support the XPO6/RAN complex, as siRNA-mediated knockdown of RASSF1A decreased association between XPO6 and RAN (Fig 3C). Expression of RASSF1A in MDA-MB-231 cells significantly enhances the association of XPO6 with RAN (Fig EV3A). We validated this requirement for RASSF1A with a GST pull-down assay using recombinant GST-RAN and lysates from siCTRL or siRASSF1A-transfected HeLa cells (Fig EV3B). Notably, XPO6 co-immunoprecipitation (IP) indicated that the XPO6/RAN complex with RASSF1A also includes MST2, suggesting potential recruitment to the NE via the RASSF1A-MST2 interaction (Fig 3C). Depletion of MST2 expression using siRNA did not affect XPO6/RAN as dramatically as siRASSF1A, but did reduce XPO6/RASSF1A, which we believe implies that

RASSF1A may be required for stabilising XPO6/RAN at the NE, i.e. in an MST2-dependent manner, but the nucleoplasmic XPO6/RAN pool may be less dependent on RASSF1A (Fig 3C). This is supported by the fact that the RASSF1A interaction with XPO6/RAN is also dependent on MST2 and therefore NE localisation (Fig 3D). To verify this mechanism, we explored MST2 associated proteins by IP and found that XPO6/RAN interaction with MST2 was RASSF1A-dependent whereas the RASSF1A/RAN interaction with MST2 did not require XPO6 (Fig EV3C), confirming our hypothesis that XPO6/RAN complex is stabilised by MST2/RASSF1A interaction. Taken together, our results show interaction of XPO6 with RAN can occur independently of RASSF1A, but a pool of XPO6/RAN is stabilised by RASSF1A in a MST2-dependent manner at the NE.

## RASSF1A is involved in actin and profilin nuclear export process

XPO6 mediates specifically the export of actin and profilin complexes out of the nucleus (Stuven *et al*, 2003). We next determined whether RASSF1A plays a role in the XPO6-dependent nuclear export. Interestingly, subcellular nuclear/cytoplasmic fractionation of HeLa cell lysates clearly demonstrated elevated levels of profilin and actin in the nucleus upon siRNA-mediated silencing of RASSF1A, compared to cells treated with control siRNA, suggesting an impaired nuclear export (Fig 4A). To further validate these results, increase in nuclear actin and profilin in RASSF1A silenced cells was only rescued upon co-transfection of RASSF1A derivatives containing a SARAH domain or overexpression of XPO6 (Fig EV4A and B). Furthermore, to ensure these effects were specific to RASSF1A, we restored nuclear actin and profilin export with siRNA-resistant FLAG-RASSF1A (Fig EV4C). Nuclear retention of profilin in HeLa cells treated with siRNA against RASSF1A was further demonstrated by immunofluorescence (Fig 4B). Remarkably, silencing of MST2 led to altered nuclear levels of actin and profilin similar to RASSF1A depletion, indicating that the NE localisation of RASSF1A is important for regulation of XPO6 (Fig EV4D). Importantly, *RASSF1A* gene silencing does not affect the expression levels of XPO6 (Fig EV4E). To rule out the possibility that nuclear actin accumulation resulted from impaired nuclear import receptor levels, we next examined protein expression of importin-9 (IPO9), which is required for actin translocation to the nucleus (Dopie *et al*, 2012), in the absence of RASSF1A. Western blotting showed no change in the expression of IPO9 upon depletion of RASSF1A, compared with the siCTRL-transfected cells (Fig EV4E). We next wondered whether the increased nuclear actin exists in monomeric (G-actin) or filamentous state (F-actin). We visualised nuclear F-actin using 568-conjugated phalloidin and G-actin using 488-conjugated DNase I. Although we did not detect any difference in the

F-actin between siCTRL- and si-RASSF1A-treated cells, we observed that significant increase in G-actin in RASSF1A-depleted cells (Figs 4C and EV4F). The significance of RASSF1A for XPO6-

mediated export process is also evident in MDA-MB-231 cells that lack RASSF1A expression. Strikingly, forcing the expression of *RASSF1A* either by transiently transfecting MDA-MB-231 cells with

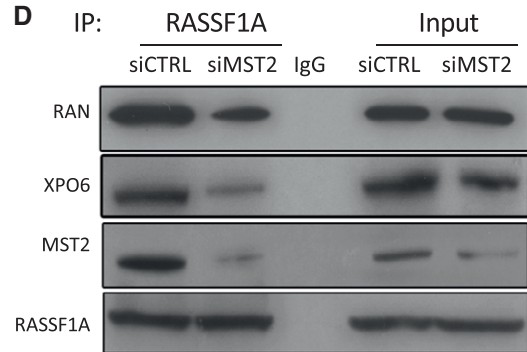

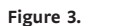

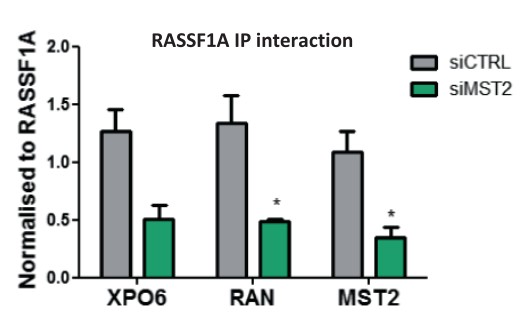

**Figure 3.**

**Figure 3. RASSF1A binds to XPO6 though its SARAH domain.**

A Co-immunoprecipitation of endogenous RASSF1A with endogenous RAN, CRM1/XPO1, XPO4, XPO5, XPO6 and XPO7 from HeLa cell lysates, compared with the IgG control.

B *Upper*: graphical representation of the domain structure of full-length RASSF1A and mutant constructs used for mapping RASSF1A/XPO6 interaction. Different domains are abbreviated as follows: SARAH, Salvador-RASSF-Hippo domain; C1, N-terminal C1 type zinc fingers; and RA, Ras-binding domain. *Lower*: Western blot analysis of MYC immunoprecipitation from the indicated inputs from HeLa cells. Co-immunoprecipitation of endogenous XPO6 with RAN in siRASSF1A from HeLa cell lysates.

C Co-immunoprecipitation of endogenous XPO6 with endogenous RAN, RASSF1A and MST2 in siRASSF1A and siMST2 HeLa cells. Quantification of the interaction of XPO6 with RASSF1A, RAN and MST2 relative to XPO6 is shown. Error bars derive from three independent experiments and represent the SEM.

D Co-immunoprecipitation of endogenous RASSF1A with endogenous XPO6 and RAN in siRNA-mediated knockdown of MST2 HeLa cells. Quantification of the interaction of RASSF1A with XPO6, RAN and MST2 relative to RASSF1A is shown. Error bars derive from three independent experiments and represent the SEM.

Data information: Two-tailed Student's *t*-test was used for statistical analysis. *$P < 0.05$, **$P < 0.01$.
Source data are available online for this figure.

---

a plasmid encoding *RASSF1A* or by treating cells with the demethylating agent 5′-aza-dC led to a decrease in actin and profilin nuclear retention (Figs 4D and EV4G). Taken together, these data strongly indicate that RASSF1A governs the export of actin and profilin out of the nucleus.

**Loss of RASSF1A expression alters MRTF-A/SRF axis**

Previous studies showed that nuclear actin has been linked to gene transcription (Visa & Percipalle, 2010; Grosse & Vartiainen, 2013; Kapoor & Shen, 2014), downregulating the expression of *MYL9*, *ITGB1* and *PAK1* (Sharili *et al*, 2016). Therefore, we measured mRNA levels of these nuclear actin-regulated genes and show that these genes were significantly reduced following siRNA-mediated knockdown of RASSF1A in comparison with control (siCTRL) HeLa cells (Fig 5A). Furthermore, mRNA encoding the transcription factor *OCT4*, which is known to be activated by nuclear actin (Yamazaki *et al*, 2015), was elevated in HeLa cells depleted of RASSF1A (Fig 5A). It is well established that depletion of IPO9 inhibits nuclear import of actin (Dopie *et al*, 2012). As expected, the levels of actin in nuclear extracts from HeLa cells treated with siRNA against IPO9 were decreased and prevented accumulation even in the absence of export via RASSF1A silencing (Fig EV5B). Moreover, the levels of *MYL9*, *ITGB1*, *PAK1* and *OCT4* mRNA upon RASSF1A silencing were rescued in HeLa cells co-depleted of IPO9 (Fig 5A).

We next tested whether increased nuclear actin levels arising from RASSF1A depletion have functional consequences for the cells. Since multiple nuclear actin-regulated genes encode for known regulators of cell adhesion (Sharili *et al*, 2016), we then assessed whether these cellular functions are affected in RASSF1A-depleted cells. In accordance with our qRT–PCR data, we showed that HeLa cells lacking RASSF1A exhibited a significantly decreased number of adhesive cells (Fig 5B). Accordingly, restoring actin levels in the nucleus by silencing IPO9 in the absence of RASSF1A was sufficient to rescue this effect (Fig 5B). Collectively, our data show that RASSF1A has an indirect effect on the expression of genes involved in cell adhesion processes, via regulation of actin levels within the nucleus.

Nuclear actin plays a key role in the regulation of the localisation and activity of myocardin-related transcription factor A (MRTF-A), a co-activator of the transcription factor serum response factor (SRF), which regulates the expression of many cytoskeletal genes (Sotiropoulos *et al*, 1999; Miralles *et al*, 2003;

Vartiainen *et al*, 2007; Ho *et al*, 2013). We next hypothesised that both MRTF-A localisation and transcriptional activity of SRF could be affected by RASSF1A protein levels. Notably, we observed that MRTF-A resides mostly in the cytoplasm in the absence of RASSF1A, whereas it is located in both nucleus and cytoplasm in control cells (Fig 5C). These observations are in line with our findings showing increased monomeric G-actin levels in RASSF1A-depleted cells as G-actin binds to MRTF-A and promotes its nuclear export (Miralles *et al*, 2003; Vartiainen *et al*, 2007). In agreement with these studies, we also showed an increased interaction between actin and MRTF-A in siRASSF1A-treated cells (Fig EV5C). Furthermore, increased nuclear G-actin levels exclude MRTF-A from the nucleus and block SRF-dependent gene transcription (Posern *et al*, 2002; Vartiainen *et al*, 2007). Accordingly, we showed decreased mRNA levels of the *SRF* gene together with decreased FBS-stimulated SRF reporter activity in cells transfected with siRNA against RASSF1A (Figs 5D and EV5D). Silencing IPO9 restored MRTF-A nuclear localisation and SRF expression in RASSF1A-depleted cells (Fig 5C and D). These results demonstrate a correlation between RASSF1A expression and SRF axis regulation, via actin localisation.

**Correlation of RASSF1 and SRF expression in human tumours**

We next analysed *SRF* gene expression from clinical data available from The Cancer Genome Atlas (TCGA) database with the cBioPortal tool (http://www.cbioportal.org). *RASSF1A* promoter methylation is widely appreciated to correlate with adverse prognosis, and we find a significant correlation between this epigenetic event and *SRF* mRNA levels indicating that loss of nuclear actin regulation and MRTF-A/SRF is likely to contribute to clinical parameters (Fig 6A, top). In line with a role in tumour suppression, *SRF* levels are significantly reduced in invasive breast cancer compared with control normal tissues (Wilcoxon $P = 1.164e^{-20}$; Fig 6B, bottom). Additionally, breast invasive carcinomas (TCGA) display higher *SRF* mRNA expression in tumours with high levels of *RASSF1A* transcript ($P < 0.0001$), which also shows significant linear correlation across the dataset ($R = 0.28$; Fig 6B). We find a matching association of *RASSF1A* and *SRF* in hepatocellular carcinoma (TCGA; $P < 0.0001$), again with significant linear correlation of *RASSF1/SRF* transcripts ($R = 0.29$; Fig 6C) and further correlations with bladder and colorectal cancer (Fig EV6). Notably, *SRF* mRNA levels were enriched in the RASSF1[high] individuals in four distinct cancer types (Figs 6 and EV6). Finally, we examined the breast and liver cancer

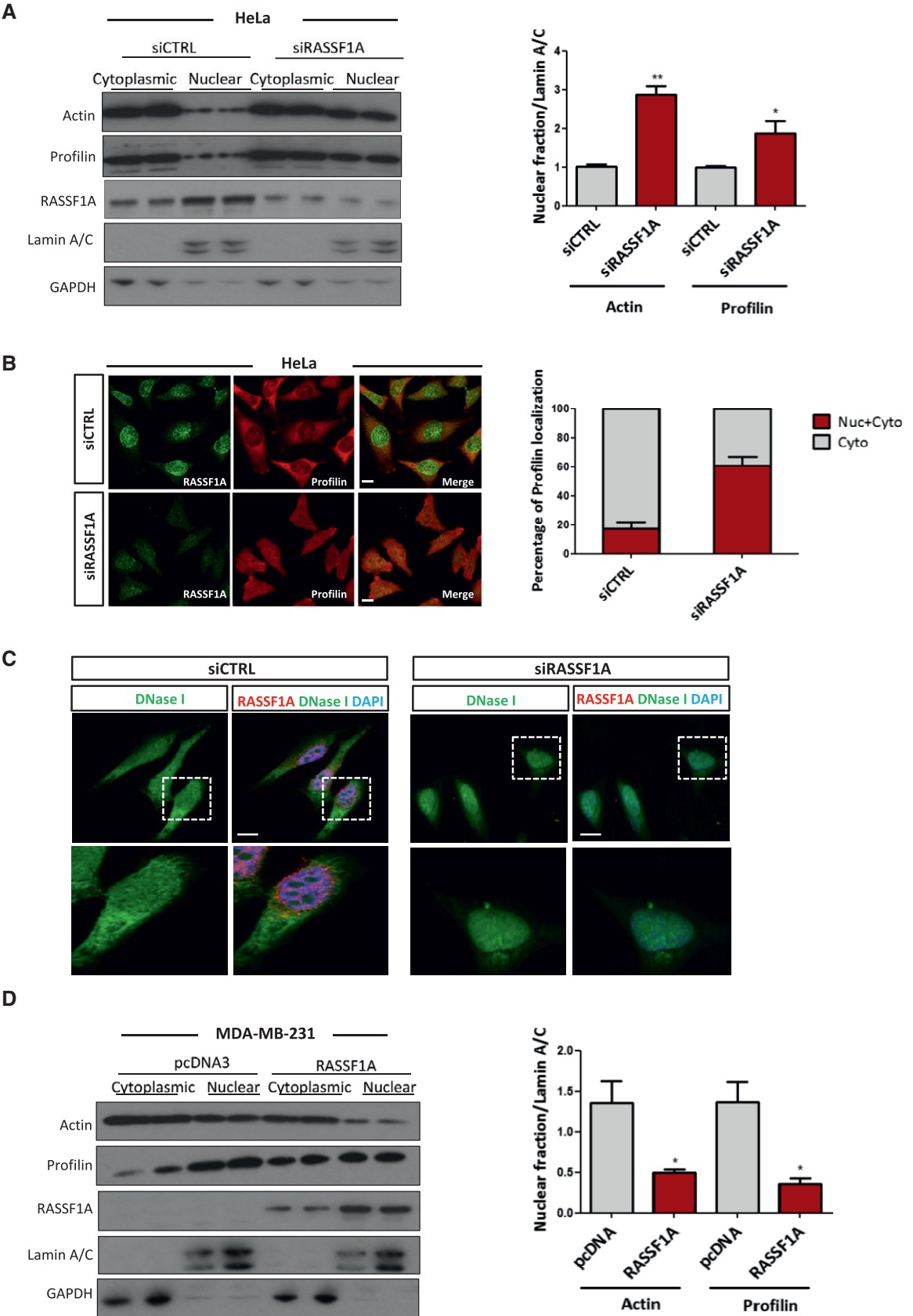

**Figure 4.**

**Figure 4. RASSF1A is involved in actin and profilin nuclear export process.**

A    HeLa cells treated with control or *RASSF1A* siRNA were fractionated into cytoplasmic and nuclear extracts. Lysates from each fraction were probed for actin and profilin alongside GAPDH (as a marker of the cytoplasmic fraction) and Lamin A/C (as a marker of the nuclear fraction). *Right:* quantification of nuclear actin and profilin relative to Lamin A/C is shown. Error bars derive from two independent experiments and represent the SEM.

B    Immunofluorescence images of profilin in control and RASSF1A siRNA-transfected HeLa cells. *Right:* the profilin localisation was scored as nuclear/cytoplasmic or predominantly cytoplasmic in approximately 100 cells. Error bars derive from three independent experiments and represent the SEM.

C    Confocal images of endogenous monomeric globular actin (G-actin) in siCTRL and siRASSF1A cells using DNase I staining (Alexa Fluor 488-conjugated, *green*). Scale bars = 10 µm.

D    Western blot analysis of actin, profilin, GAPDH and Lamin A/C levels in nuclear and cytoplasmic fractions of MDA-MB-231 cells treated with control pcDNA3 or *RASSF1A* vector. The graph shows the nuclear levels of actin and profilin relative to Lamin A/C in MDA-MB-231 cells expressing RASSF1A. Error bars derive from two independent experiments and represent the SEM.

Data information: Two-tailed Student's *t*-test was used for statistical analysis. *$P < 0.05$, **$P < 0.01$.
Source data are available online for this figure.

survival statistics in the TCGA and we found that patients with low SRF expression have a poor prognosis for overall survival (Breast HR = 0.8 (0.72–0.89) Logrank $P = 5e^{-5}$; HCC HR = 0.67 (0.47–0.96) Logrank $P = 0.027$; Fig 6D). Overall, our data show a correlation of *RASSF1* and *SRF* mRNA expression and SRF levels associate with poor survival.

## Discussion

Actin constantly shuttles between the cytoplasm and the nucleus using an active transport mechanism, and proper balance of nuclear and cytoplasmic actin pools is tightly regulated. Nuclear actin is exported from the nucleus by XPO6, independently of the general export receptor CRM1, and it is imported by IPO9 (Stuven *et al*, 2003; Dopie *et al*, 2012). As for most export processes, XPO6 also requires RAN GTPase to export actin monomers in complex with profilin from the nucleus (Stuven *et al*, 2003). We found that lack of RASSF1A, a phenomenon commonly observed in human tumours, leads to accumulation of actin within the nucleus due to defective nuclear export. The direct interaction of RASSF1A with XPO6, a member of the importin-β superfamily of transport receptors, is required for the association with RAN. Intriguingly, the RASSF1A/RAN/XPO6 complex includes MST2, suggesting that the recruitment of RASSF1A to MST2 at the NE potentially supports XPO6 binding via the SARAH domain (Dittfeld *et al*, 2012) and this is important for the active participation in actin-profilin nucleocytoplasmic shuttling. However, the reduced involvement of RASSF1A in the XPO6/RAN complex in the absence of MST2 suggests that RAN/XPO6 exists independently of RASSF1A in the nucleoplasm. This means that XPO6/RAN complexes may be contextually distinct from RASSF1A/RAN/XPO6 and could involve differences in substrate loading, RAN GDP/GTP loading or post-translational modifications of RAN (Dallol *et al*, 2009; Bompard *et al*, 2010; Güttler & Görlich, 2011; de Boor *et al*, 2015).

Reduction in cytoplasmic actin was not evident as a consequence of the defective export, most likely because of the higher actin concentration in the cytoplasm (Fig 4A). Nuclear actin is a key regulator of transcription, and it is required for all three RNA polymerases (Pol I, II and III; Philimonenko *et al*, 2004; Qi *et al*, 2011). Previous reports showed that expression of nuclear actin negatively regulates multiple genes and results in altered expression of approximately 2,000 genes (Yamazaki *et al*, 2015; Sharili *et al*, 2016). Loss of *RASSF1A* expression alters gene expression

and impacts cellular processes, such as cell adhesion. Thus, RASSF1A appears to contribute to the transcriptional activity of the cell and further to its phenotypic changes through its ability to modulate nuclear actin levels. Although nuclear actin levels have not been extensively studied in cancer cells, abnormal subcellular localisation of certain ABPs (actin-binding proteins) is associated with the development carcinogenesis processes (Loy *et al*, 2003; Velkova *et al*, 2010; Khurana *et al*, 2011; Savoy & Ghosh, 2013; Honda, 2015; Patnaik *et al*, 2016). A recent study showed that the nuclear actin levels regulated by XPO6 are pivotal for cell proliferation and quiescence and that preventing actin export from the nucleus contributes to malignant progression (Fiore *et al*, 2017). Of note, low XPO6 expression and therefore nuclear actin accumulation correlate with poor survival of breast cancer patients (Fiore *et al*, 2017).

Mechanotransduction signalling is essential for a broad range of biological functions such as embryogenesis, cell migration, metastasis and epithelial–mesenchymal transition. MRTF-A is a mechanosensitive transcription co-factor which, together with SRF, controls a variety of genes involved in actin cytoskeletal remodelling and growth factor response. MRTF-A/SRF activation is responsible for immediate transcriptional response to mechanical stimuli (Cui *et al*, 2015) and perturbed MRTF-A/SRF signalling axis might partially explain the pathogenesis of certain human diseases (Ho *et al*, 2013). We found that the SRF-driven gene transcription activity in response to serum stimulation was severely abrogated in RASSF1A-depleted cells (Fig 5D). In addition, we also found that *RASSF1* expression correlates with *SRF* expression in a variety of cancers (Fig 6). Although there is currently lack of evidence, altered MRTF-A/SRF signalling may explain the aberrant mechanosensitivity observed often in cancer cells (Ghosh *et al*, 2008; Tang *et al*, 2012; Ciasca *et al*, 2016).

Epigenetic silencing of the *RASSF1A* gene has been frequently associated with poor patient outcome across all solid malignancies (Grawenda & O'Neill, 2015). Several studies have identified functional roles for RASSF1A-mediated tumour suppression that can explain these clinical associations, e.g. Hippo pathway regulation (Matallanas *et al*, 2007; Hamilton *et al*, 2009), apoptosis (Vos *et al*, 2000; Baksh *et al*, 2005), differentiation (Papaspyropoulos *et al*, 2018), the cytoskeleton (Liu *et al*, 2003; Dallol *et al*, 2004; Vlahov *et al*, 2015) and DNA damage/repair (Pefani *et al*, 2014, 2018; Donninger *et al*, 2015). However, the widespread prognostic associations imply a perturbation of a biological process that simultaneously contributes to these diverse

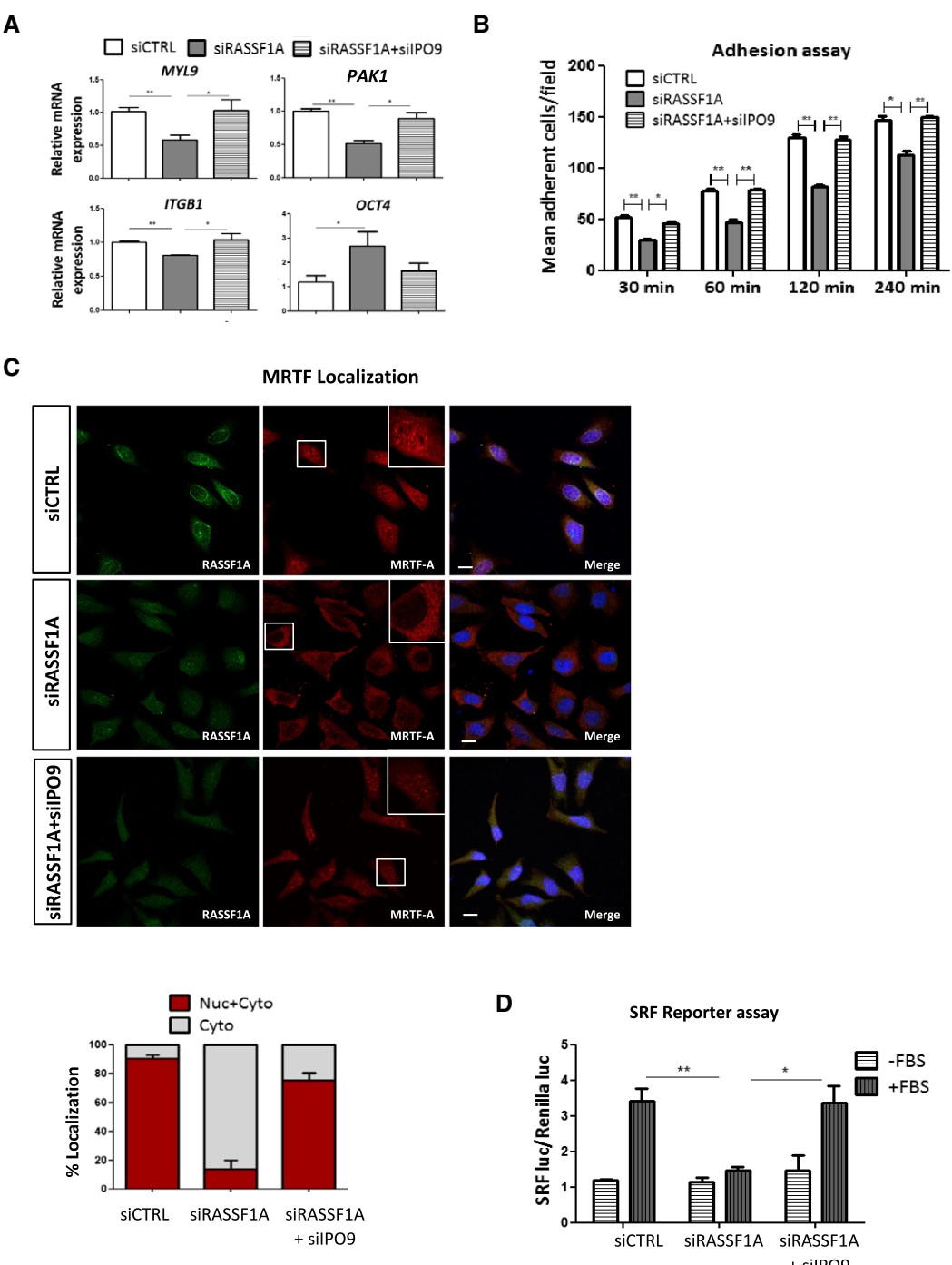

**Figure 5. Loss of RASSF1A expression alters MRTF-A/SRF axis.**

A  qRT–PCR validation of selected genes known to be affected by the levels of nuclear actin. Transcript levels of *MYL9*, *ITGB1*, *PAK1* and *OCT4* from HeLa cells treated either with siRASSF1A or with siRASSF1A + siIPO9 are relative to *GAPDH* and normalised to siCTRL cells. Data represent SEM of three independent experiments.

B  Adhesion assay. siRNA against *RASSF1A* significantly decreased HeLa cells' adhesive rate at all the determined time points compared with control. The cells were cultured for 48 h before harvesting and reseeding for 1 h on 96-well plates coated with FN. Data represent the SEM of three independent experiments.

C  Representative immunofluorescence images of MRTF-A localisation in siRASSF1A and siRASSF1A + siIPO9. *Lower:* the MRTF-A localisation was scored as nuclear/cytoplasmic or predominantly cytoplasmic in 100–200 cells. DNA was stained with DAPI. Error bars derive from two independent experiments and represent the SEM. Scale bars = 10 μm.

D  Luciferase assay of SRF-dependent promoter in cells transfected with siCTRL, siRASSF1A or siRASSF1A + siIPO9 following stimulation with 10% FBS for 5 h. Data are expressed as SRF luciferase activity relative to Renilla control and represent the SEM of two independent experiments.

Data information: Two-tailed Student's *t*-test was used for statistical analysis. *P < 0.05, **P < 0.01.

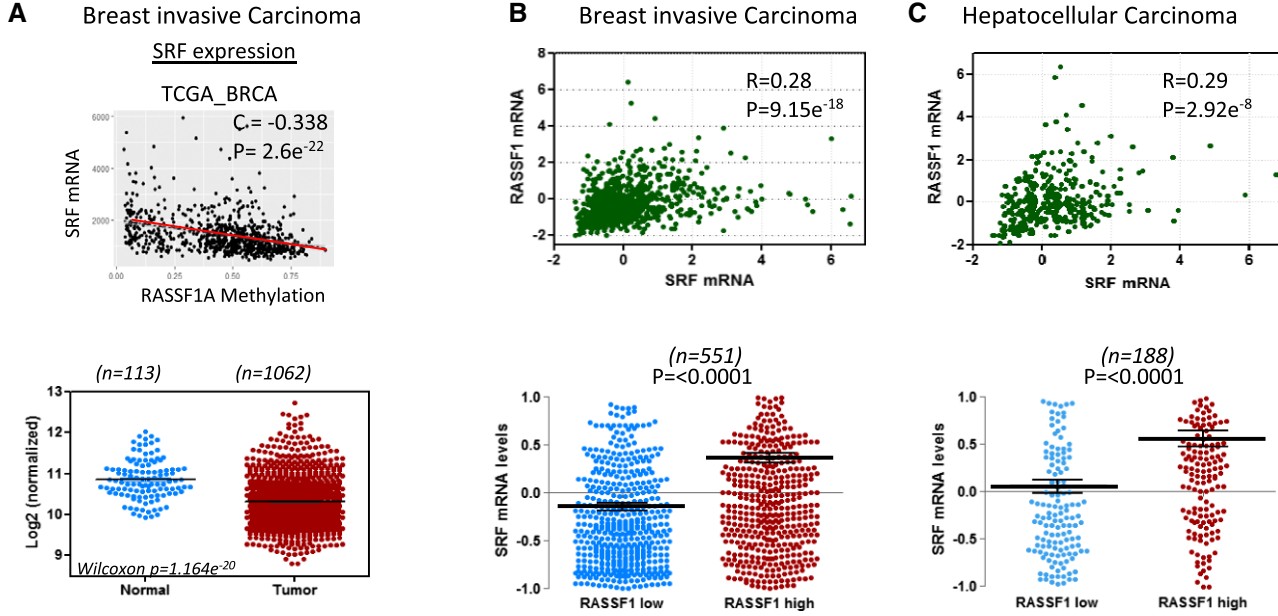

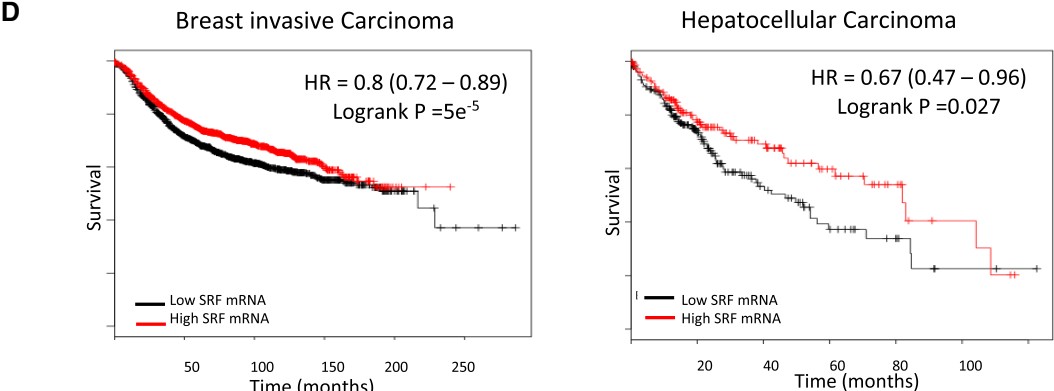

**Figure 6. Correlation of RASSF1 and SRF expression in human tumours.**

A  Upper: scatter plot of RASSF1 methylation levels to SRF mRNA expression in breast invasive carcinoma (TCGA, Provisional, *n* = 960). Lower: SRF expression (TCGA, Illumina HiSeq RNA Seq) in normal and BRCA patients.

B  Upper: scatter plot of RASSF1 mRNA to SRF mRNA levels in breast invasive carcinoma (TCGA, Provisional, *n* = 960). Values are given in (RNA Seq V2 RSEM). Lower: differences in SRF mRNA expression of samples that express low and high levels of RASSF1 mRNA.

C  Upper: scatter plot of RASSF1 mRNA to SRF mRNA levels in hepatocellular carcinoma (TCGA, Provisional, *n* = 360 samples). Values are given in (RNA Seq V2 RSEM). Lower: differences in SRF mRNA expression of samples that express low and high levels of RASSF1 mRNA.

D  Survival analysis of BRCA (*n* = 3,951) and HCC (*n* = 364) patient samples estimated by Kaplan–Meier survival curve with high (red) and low (black) expression of SRF.

mechanisms, but this remains unexplained. Intriguingly, nuclear actin levels have also been described to contribute to the regulation of Hippo pathway (e.g. SRF regulation of YAP; Sen *et al*, 2015; Foster *et al*, 2017), apoptosis (Sharili *et al*, 2016), differentiation (Xu *et al*, 2010; Sen *et al*, 2015, 2017) and DNA damage/ repair (Yuan & Shen, 2001; Belin *et al*, 2015). Altogether, our findings indicate that loss of RASSF1A expression results in failure to export nuclear actin, suggesting that both regulatory processes are linked and that the clinical data associated with *RASSF1* methylation involve deregulated MRTF-A/SRF.

# Materials and Methods

### Tissue culture and cell treatments

HeLa, MEFs, HEK293, RPE-1 and MDA-MB-231 cells were cultured in complete DMEM supplemented with 10% foetal bovine serum in 5% $CO_2$ and 20% $O_2$ at 37°C. HeLa cells were purchased from Cancer Research UK, London, or LGC Promochem (ATCC). Inhibitors for ATR kinase activity (VE-821) and ataxia-telangiectasia-mutated (ATM) kinase activity (KU-55933) were used at a

concentration of 10 μM. 10 μM 5-aza-2′-deoxycytidine (5′-aza-dC; Sigma) was added to the medium, and the cells were incubated for 72 h. Due to its chemical instability, 5′-aza-dC was added to the fresh medium every 24 h.

## Real-time PCR primers

| Target gene | Forward | Reverse |
| --- | --- | --- |
| PAK1 | ACCACCAGTGATTGCTCCAC | GCATCTGGTGGAGTGGTGTT |
| ITGB1 | TCCAACCTGATCCTGTGTCC | CAATTCCAGCAACCACACCA |
| MYL9 | GTTTGGGGAGAAGCTGAACG | CCGGTACATCTCGTCCACTT |
| GAPDH | AACGGGAAGCTTGTCATCAA | CCCAGCCTTCTCCATGGTG |
| OCT4 | GGTCCGAGTGTGGTTCTGTA | CGAGGAGTACAGTGCAGTGA |
| SRF | GGAGACCAAGGACACACTGA | TGCCTGTACTCTTCAGCACA |
| IPO9 | TGGGTGAGAGCAGAAGGTCT | CTTCCTGCTGACACTGGACA |

## siRNA oligonucleotides

RNA interference was carried out with 100 nM siRNA for 48 h using Lipofectamine 2000 (Invitrogen) according to manufacturer's instructions. The oligos used were as follows: siMST2: siGENOME SMARTpool: M-004874-02 (Dharmacon), siRASSF1A: GACCUCU-GUGGCGACUU, siLaminA/C: sc35776 (Santa Cruz Biotechnology, Inc), siIPO9: EHU102331 (Sigma). siRNA against luciferase with the sequence GCCAUUCUAUCCUCUAGAGGAUG was used as control.

## Plasmids

RASSF1A truncation mutants, MYC-RASSF1A 1–288, MYC-RASSF1A 120–340 and MYC-RASSF1A120–288, were created by PCR using MYC-tagged RASSF1A as template, as previously shown (Pefani et al, 2016). MDA-MB-231 cells were transiently transfected with 1 μg of RASSF1A DNA or the empty pcDNA 3.0 plasmid using Lipofectamine 2000 (Invitrogen).

## Antibodies

The following antibodies were used in this study: RASSF1A (Atlas, HPA040735), MST1 (Sigma, WH0006789M1), MST2 (Abcam, ab71960, ab52641), XPO6 (Bethyl, A301-205A), CRM1 (Cell Signaling, 46249), XPO4 (Novus Biologicals, NB100-56495), XPO5 (sc-166789), XPO7 (sc-390025), Lamin A/C (Cell Signaling, 4777), Lamin B (Abcam, ab16048), Profilin (Novus Biologicals, NBP2-02577), IPO9 (Abcam, ab52605), Myc-Tag (JBW301, Millipore, 05-724), GAPDH (Cell Signaling, 97166), Actin (sc-1616, Santa Cruz), RAN (Santa Cruz Biotechnology, sc58467), MRTF-A (Santa Cruz Biotechnology, sc-398675), α-tubulin (Abcam, ab7291) and pS131-RASSF1A custom-made (Hamilton et al, 2009).

## Quantitative real-time PCR analysis

RNA extraction, reverse transcription and qPCR were performed using the Ambion Power SYBR Green Cells-to-CT kit following manufacturer's instructions in a 7500 FAST Real-Time PCR thermocycler with v2.0.5 software (Applied Biosystems). The mRNA expression level of each gene relative to GAPDH was calculated using the $\Delta\Delta C_t$ method. All experiments were performed in triplicates.

## Protein extraction and immunoblotting

Cytosolic and nuclear fractions were prepared using the NE-PER Nuclear and Cytosolic Extraction Reagents (Thermo Fisher Scientific) according to the manufacturer's instructions. Sample protein content was determined by the bicinchoninic acid assay protein assay (Thermo Fisher Scientific). Extracts were analysed by SDS–PAGE using a 10% Bis–Tris NuPAGE gels (Invitrogen) and transferred onto PVDF membranes (Millipore). Subsequent to being washed with PBS containing 1% Tween-20 (PBS-T), the membranes were blocked in 5% bovine serum albumin (BSA) in TBS-T for 1 h at RT and then incubated with the primary antibody overnight at 4°C. The membranes were incubated with HRP-conjugated secondary antibodies for 1 h at room temperature and exposed to X-ray film (Kodak) after incubation with Thermo Scientific Pierce ECL or Amersham ECL (GE Healthcare). ImageJ software (NIH) was used for the quantification of the bands. All bands were normalised against the loading controls.

## Immunofluorescence

HeLa cells were grown on coverslips, washed with PBS and fixed with MeOH at 20°C for 20 min. Where indicated, cells were instead fixed in 4% paraformaldehyde and permeabilised either with 0.5% Triton X-100 at room temperature for 5 min or with 40 μg/ml digitonin on ice for 2 min. Coverslips were then incubated with primary antibody for 2 h in PBS with 3% BSA at room temperature. Cells were washed with PBS and incubated with secondary anti-rabbit and/or anti-mouse or anti-rat IgG conjugated with Alexa Flour 488 or Alexa Flour 568 (Molecular Probes) for 1 h at room temperature. Samples were washed and mounted on coverslips with mounting medium containing DAPI (Invitrogen). For the visualisation of F- and G-actin, cells were fixed with 4% paraformaldehyde, permeabilised with 100% acetone at 20° for 5 min and stained with Alexa Fluor 568 Phalloidin or Alexa Fluor 488 DNase I, respectively. Cells were analysed using LSM780 (Carl Zeiss Microscopy) confocal microscope. Fluorescence intensity plots were generated using ImageJ software (NIH).

## Immunoprecipitation

Cells were lysed in lysis buffer [50 mM Tris–HCl pH 7.5, 0.15 M NaCl, 1 mM EDTA, 1% NP-40, 1 mM $Na_3VO_4$, complete proteinase inhibitor cocktail (Roche)] and incubated with 20 μl protein G Dynabeads (Invitrogen) and 2 μg of the indicated antibodies at for 2 h at 4°C. Pelleted beads were collected in sample buffer NuPAGE LDS (Thermo Fisher Scientific) with 200 mM DTT and subjected to SDS–PAGE and immunoblotting.

## GST pull-down assay

HeLa cells transfected with siCTRL or siRASSF1A were lysed in lysis buffer (25 mM Tris–HCl, pH 7.2, 150 mM NaCl, 5 mM $MgCl_2$, 1%

NP-40 and 5% glycerol). 20 μg of GST-RAN was mixed with 500 μg of lysates in spin cups (Thermo Fisher Scientific #69700) containing 40 μl of 50% glutathione-agarose beads (Thermo Fisher Scientific #16100). The samples were rotated for at least 1 h at 4°C. The beads were washed five times with the lysis buffer, mixed with 40 μl of 2× Laemmli buffer and boiled for 10 min at 95°C. The beads were pelleted and the supernatant used for analysis via SDS–PAGE and Western blot. GST-RAN recombinant protein was purchased from Novus Biologicals.

## Mass spectrometry

HeLa cells were lysed in 1% NP-40 lysis buffer (150 mM NaCl, 20 mM HEPES, 0.5 mM EDTA) containing complete protease and phosphatase inhibitor cocktail (Roche). 10 mg of total protein lysate was incubated for 3 h with protein A Dynabeads (Invitrogen) and 10 μg of MST2 antibody (ab52641) or rIgG at 4°C. MST2 immunoprecipitates were eluted off the beads in a low pH glycine buffer. The eluted fractions were sequentially incubated with DTT (5 mM final concentration) and Iodoacetamide (20 mM final concentration) for 30 min at room temperature in the dark, before proteins were precipitated with methanol/chloroform. Protein precipitates were reconstituted and denatured with 8 M urea in 20 mM HEPES (pH 8). Samples were then further diluted to a final urea concentration of 2 M using 20 mM HEPES (pH 8.0) before adding immobilised trypsin for 16 h at 37°C (Pierce 20230; Montoya *et al* 2011). Trypsin digestion was stopped by adding 1% TFA (final concentration) and trypsin beads removed by centrifugation. Tryptic peptides solution was desalted by solid-phase extraction using C18 Spin Tips (Glygen LTD) and dried down.

Dried tryptic peptides were reconstituted in 15 μl of LC-MS grade water containing 2% acetonitrile and 0.1% TFA. Seven per cent of the sample was analysed by liquid chromatography–tandem mass spectrometry (LC_MS/MS) using a Dionex UltiMate 3000 UPLC coupled to a Q-Exactive mass spectrometer (Thermo Fisher Scientific). Peptides were loaded onto a trap column (PepMap C18; 300 μm × 5 mm, 5 μm particle size, Thermo Fisher) for 1 min at a flow rate of 20 μl/min before being chromatographic separated on a 50-cm-long Easy-Spray Column (ES803, PepMAP C18, 75 μm × 500 mm, 2 μm particle, Thermo Fisher) with a gradient of 2–35% acetonitrile in 0.1% formic acid and 5% DMSO with a 250 nl/min flow rate for 60 min (Chung *et al* 2016). The Q-Exactive was operated in a data-dependent acquisition (DDA) mode to automatically switch between full MS scan and MS/MS acquisition. Survey-full MS scans were acquired in the Orbitrap mass analyser over an m/z window of 380–1,800 and at a resolution of 70 k (AGC target at 3e6 ions). Prior to MSMS acquisition, the top 15 most intense precursor ions (charge state ≥ 2) were sequentially isolated in the Quad (m/z 1.6 window) and fragmented in the HCD collision cell (normalised collision energy of 28%). MS/MS data were obtained in the Orbitrap at a resolution of 17,500 with a maximum acquisition time of 128 ms, an AGC target of 1e5 and a dynamic exclusion of 27 s.

The raw data were searched against the Human UniProt–SwissProt database (November 2015; containing 20,268 human sequences) using Mascot data search engine (v2.3). The search was carried out by enabling the Decoy function, whilst selecting trypsin as enzyme (allowing 1 missed cleavage), peptide charge of +2, +3, +4 ions, peptide tolerance of 10 ppm and MS/MS of 0.05 Da; #13C at 1; carbamidomethyl (C) as fixed modification; and oxidation (M) and deamidation (NQ) as a variable modification. MASCOT outputs were filtered using an ion score cut-off of 20 and a false discovery rate (FDR) of 1%.

## Cell adhesion assay

HeLa cells were transfected with siRNA against RASSF1A or RASSF1A and IPO9 or control and were collected after 48 h. $1 \times 10^5$ cells were re-plated on fibronectin-coated 24-well plate in triplicate. After attachment for the predetermined time (30, 60 and 90 min), the plate was washed to remove the non-adherent cells. The number of adherent cells from four random fields of the well was counted microscopically.

## SRF reporter assay

Cells were transfected with the SRF luciferase reporter (pGL4.34) using Lipofectamine 2000. pRL Renilla luciferase vector was used as an internal control. Following DNA transfection, cells were rinsed and cultured for 24 h before treating with FBS. Luciferase assays were carried out in 24-well plates (*n* = 3 wells); cells were treated for 7 h with FBS, then harvested and analysed using the dual luciferase assay (Promega).

## Correlation analysis

The TCGA data were downloaded from the cBioPortal for Cancer Genomics (http://www.cbioportal.org). The association of RNA expression of RASSF1 and SRF was studied in breast invasive carcinoma (TCGA, Provisional), hepatocellular carcinoma (TCGA, Provisional), bladder cancer (Robertson *et al* 2017) and colorectal adenocarcinoma (Cancer Genome Atlas Network 2012). Methylation 450K data and gene expression data were downloaded from TCGA (BRCA). RASSF1A methylation levels for each patient were determined as the average beta value across all probes within the RASSF1A CpG island (chr3:50377804-50378540). Pearson correlation and scatter plots were drawn using R (Ver 3.4). Survival analyses from 3951 BRCA samples and 364 liver HCC patient samples were performed using the Kaplan–Meier Plotter tool with the patients being split by median (Nagy *et al*, 2018).

## Statistics

Statistical analysis was performed with the GraphPad Prism 7 software. For all experiments, statistical analysis was carried out using Student's *t*-test. All data are expressed as mean ± SEM.

# Data availability

The mass spectrometry raw data included from this publication have been deposited to the ProteomeXchange Consortium via the PRIDE partner repository and assigned the identifier PRIDE: PXD11517 (https://www.ebi.ac.uk/pride/archive/projects/PXD11517).

**Expanded View** for this article is available online.

## Acknowledgements

This work was funded by Cancer Research UK A19277. Dafni Eleftheria Pefani is funded by the Hellenic Foundation for Research and Innovation (HFRI) and

the General Secretariat for Research and Technology (GSRT), under grant agreement No [775]. Mass spectrometry analysis was performed at the Discovery Proteomics Facility (headed by Roman Fischer) which is part of the TDI MS Laboratory (led by Benedikt Kessler).

## Author contributions

MC conceived the project and designed the research, performed experiments and analysed data. D-EP, DP and ME assisted with experiments. IV and RF performed the mass spectrometry analysis. EON and MC wrote the manuscript.

## Conflict of interest

The authors declare that they have no conflict of interest.

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
