## [Review Process File · The EMBO Journal]

RASSF1A IS REQUIRED FOR THE MAINTENANCE OF NUCLEAR ACTIN LEVELS

Maria Chatzifrangkeskou, Dafni-Eleftheria Pefani, Michael Eyres, Iolanda Vendrell, Roman Fischer, Daniela Pankova, and Eric O'Neill

Review timeline:

Submission date:	19th Nov 2018
Editorial Decision:	20th Dec 2018
Revision received:	20th Mar 2019
Editorial Decision:	17th Apr 2019
Revision received:	23rd Apr 2019
Accepted:	14th May 2019

Editor: Ieva Gailite

Transaction Report:

1st Editorial Decision

20th Dec 2018

Thank you for submitting your manuscript for consideration by the EMBO Journal. We have now received three referee reports on your manuscript, which are included below for your information.

As you can see from the comments, all three referees express interest in the presented role of RASSF1A in regulation of nuclear actin export. However, they also raise substantive concerns with the analysis that need to be addressed before they can support publication here. Based on the overall interest expressed in the reports, I would like to invite you to submit a revised version of your manuscript in which you address the comments of all three referees. I would ask you to focus in particular on the following points:

- All referees request additional analysis and support for the role of RASSF1A in formation and function of the XPO6/RanGTP complex (referee #1, point 5; referee #2, point 2; referee #3, point 1)
- Further investigation of the regulation of MRTF-A by RASSF1A (Referee #1, point 6; referee #2, point 3)
- Provide further evidence for regulation nuclear actin levels upon RASSF1A manipulation (reviewer #1, points 4 and 7, reviewer #3, point 3)
- Clarify the role of MST2 for RASSF1A localisation and RASSF1A/XPO6/RanGTP complex formation (reviewer #1, point 3; reviewer #2, point 1)

I should add that it is The EMBO Journal policy to allow only a single major round of revision and that it is therefore important to resolve the main concerns raised at this stage. Since considerable additional work would be needed to fulfill all referee requests, please contact me if you would like to discuss the feasibility of any of the experiments for the revision.

REFeree REPORTS:

Referee #1:

In this study the authors identify RASSF1A, the Hippo kinase regulator, as a regulator of actin monomer levels in the nuclear compartment via interacting with exportin 6, the specific actin export factor, as well as with the Ran GTPase. This leads to changes in actin-dependent gene regulation and may have functional effects on cell adhesion, all supposedly mediated by subsequent changes in the activity of MRTF-A and SRF. Finally, the authors try to correlate their findings with clinical data of invasive carcinoma. Overall, the findings are interesting since they identify a novel player in the nuclear actin-MRTF module, however, several further controls are needed.

points of critique:

1. Fig.2C the quantification of effective-knockdowns should be included; RASSF1A decrease resulted from MST2 needs to be confirmed via Western blot.
2. It is claimed that SARAH domain is responsible for the interaction, therefore, in Fig. 3B it is necessary to rule out the involvement of RA domain by introducing a 288-340 mutation.
3. Fig. 3C as the co-immunoprecipitation assay showed, Ran recruitment to the RASSF1A/XPO6/RanGTP complex is totally abolished. Could this be seen in staining as well? The same concern also exists in the Fig. 3D, disturbed formation of the complex after MST2 depletion. Additionally, since MST2 is not involved in the RASSF1A/XPO6/RanGTP complex, how come it affects the complex formation? What is the mechanism?
4. Fig.4 is not entirely convincing. All conclusions on nuclear actin levels are based on biochemical fractionations, which can be often problematic. Besides, effects of siRASSF1A need to be rescued by siRNA resistant wt versus SARAH domain deletion derivatives! This is important and necessary. The biochemical analysis of cleat actin levels needs to be complemented by nuclear actin visualizations either with endogenous actin or at least using GFP-actin.
5. Could the effect of RASSF1A depletion on nuclear actin levels be rescued by overexpression of XPO6? This would more directly confirm XPO6 is downstream of RASSF1A. In fact, rescue and reconstitution studies would be highly desirable.
6. How does RASSF1A affect MRTF-A? The authors should study MRTF-A (also known as MAL, or MKL1!) actin interactions directly in the presence or absence of RASSF1A.
7. What happens to nuclear F-actin in RASSF1A silenced cells? Did the author make efforts to visualize endogenous nuclear F-actin? This would be interesting and provide more and better mechanistic insight, since it has been shown in several studies that nuclear actin assembly regulates MRTF-A or MKL1 or MAL activity. It is also well established by now that nuclear actin is dynamically polymerized by various cellular cues and inputs. Hence, does an increase in nuclear actin levels lead to actin polymerization or changes of nuclear F-actin in this compartment?

Referee #2:

The manuscript by Chatzifrangkeskou et al. describes a novel role for RASSF1A tumor suppressor gene in regulating nuclear export of actin, and thus also MRTF-A-SRF mediated transcription. RASSF1A is a member of a protein family acting upstream of the Hippo-pathway, binding directly to the MST1 and 2 kinases of the Hippo pathway, and its inactivation is implicated in the development of various cancers.

In the present manuscript, the authors broaden the functional aspects of RASSF1A by linking it to regulation of nuclear actin levels. Actin has been shown to shuttle in and out of the nucleus with the aid of transport factor Importin-9 and Exportin-6, but further regulators are poorly characterized. The authors first show that a pool of RASSF1 can be found on the nuclear envelope, and that this localization is dependent on MST2. Based on the previous data on the interaction between RASSF1 and the Ran-GTPase, which is the major driver of energy dependent nuclear transport, the authors then identify an interaction between RASSF1 and Exportin-6, and surprisingly find that RASSF1 is required for the interaction between Exportin-6 and Ran. In cells, depletion of RASSF1 leads to increased nuclear accumulation of actin and profilin, as well as to decreased expression of specific genes previously linked to nuclear actin, cell adhesion defects and to decreased nuclear localization of the transcription coactivator MRTF-A and consequently decreased SRF activity. Finally, the expression of SRF and RASSF1A are correlated in various cancer cells.

The outset for the manuscript is very interesting, and the idea that the nuclear actin pool, and

consequently gene expression, would be regulated by a tumour suppressor is exciting. However, the mechanism by which RASSF1A operates here remains somewhat unclear, and further experiments as well as quantifications would be needed to support the key hypothesis.

Major concern

From the data, it is clear that RASSF1A interacts with Exportin-6 and that its depletion leads to increased levels of actin in the nucleus. However, I do not completely understand the mechanism that the authors are proposing.

First, how is the NE localization of RASSF1 related to whole business? How would the NE localized RASSF1A contribute to the formation of the Exportin-6-Ran-actin export complex, especially since MST2 does not seem to be in the complex? MST2 seems to be required for RASSF1A localization to the NE, but can it be excluded that it also regulates the nuclear localization of RASSF1A? In these experiments, the subcellular localization of RASSF1A should be analyzed quantitatively from a large number of cells (or by fractionation) to draw any general conclusions. Analyzing actin distribution in MST2 depleted cells (in case it does not affect nuclear localization of RASSF1A per se) would tell, if the NE localization of RASSF1A is important for regulating nuclear actin.

Second, I am very surprised by the huge effect that RASSF1 depletion has on Exportin-6-Ran interactions (figure 3C), especially since it seems that the depletion of RASSF1 is not very efficient. Moreover, Exportin-6 has been shown to interact directly, like most other nuclear transport factors, with Ran (shown for example in Stuken et al. 2003). Why would the interaction in cells then require an additional factor? The IPs in figures 3C and D should be carefully quantified from several experiments to draw firm conclusions on these experiments. Related to this point, did the authors check the subcellular distribution of Exportin-6 in RASSF1A depleted cells?

Third, the results in figures 4C and 4D on MDA-MB-231 cells are very clear, and the data seems to imply that due to the lack of RASSF1A, these cells have relatively high levels of nuclear actin. However, the Treisman-lab has shown that in these cells, MRTF-A is actually nuclear (Medjkane et al. 2009, NCB), which is exactly the opposite that you would expect, and not in agreement with the subsequent results in figure 5. What is the explanation for this?

Fourth, it is surprising how well the Importin-9 silencing can "rescue" the effects of RASSF1A silencing on gene expression, adhesion and SRF activation in figure 5. However, key controls are missing. The authors should first of all convincingly demonstrate that Importin-9 is appropriately silenced, since the blot in S3A (of nuclear fractions only) is not very clear. They should then demonstrate that importin-9 silencing can actually "rescue" the nuclear actin levels back to normal with fractionation.

In figure 1D, the PLA experiment does add too much information, and if kept in the manuscript, should be quantified. Figure 1E would have benefitted from a positive control for the digitonin treatment (e.g. some outer nuclear membrane protein) to prove that plasma membrane is permeabilized, since the RASSF1A staining in the cytoplasm does not look very convincing.

Minor points

In quantification of figure 4C, it is not clear what the data is relative to.

At several places, the authors talk about "expression", when they should be talking about localization.

In figure 1A, the white dashed line is in the wrong place.

The materials and methods contains stuff that is not in the manuscript, e.g. on fluorouridine.

On page the sentence "Lamin A/C and B1 both contribute to NE integrity, with Lamin B constituting more stable filaments and Lamin A/C more responsible for stable rigidity." is wrong and does not contain references.

On page 7, depletion of IPO9 inhibits nuclear import of actin, NOT promotes export.

On page 7, last sentence on first paragraph does not reflect the results, since not all genes were increased.

In discussion, the first reference is not optimal, since for example the nuclear import pathway for actin was not discovered, when that paper was published.

Referee #3:

In this manuscript, the authors demonstrated that RASSF1A (Ras association domain family 1 isoform A), a tumour suppressor, functions as a novel regulator of actin nucleocytoplasmic trafficking to regulate nuclear actin levels. They found that RASSF1A localizes to nuclear envelope, and supports the binding between exportin-6 (XPO6) and RAN GTPase to modulate the nuclear actin export. Furthermore, they showed that RASSF1A is involved in the regulation of MRTF-A, a coactivator of the SRF transcription factor. This is an interesting paper containing potentially important findings and general significance. However, the current manuscript does not provide the mechanistic insight into how RASSF1A participates in nucleocytoplasmic transport process, especially, the formation of XPO6/RAN complex.

Specific comments:

(1) It has been already demonstrated that XPO6 directly binds to RAN (Q69L), using recombinant proteins (Stuven et al, 2003). So, how does RASSF1A affect the formation of XPO6/RAN complex? Since this is one of the most striking findings in this study, the authors need to clarify the role of RASSF1A.

The authors could perform a binding assay using recombinant proteins to examine how the presence of RASSF1A affects the binding between XPO6 and RAN. In addition, the authors should examine the binding between XPO6 and RAN in MDA-MB-213 cells by co-immunoprecipitation (as in Figure 3C). It may also be possible that the knockdown of RASSF1A causes the subcellular translocation of XPO6 or RAN.

(2) Figure 3C: The band intensity of XPO6 in IP sample seems too weak. Also, in Figure 3D, the band intensity of RASSF1A is weak as well. Therefore, it is not convincing if these antibodies could precipitate their antigens.

(3) Figure 4A: The effect of siRASSF1A on actin/profilin nuclear export appears to be relatively modest, considering that XPO6-RAN interaction is almost completely disrupted by the same treatment (Figure 3C). This could be due to the higher concentration of actin/profilin in cytoplasm, as the authors stated. However, to clarify this, authors should examine the effect of XPO6 knockdown on the actin/profilin nuclear export.

(4) Page 7:

"increased levels of MYL9, ITGB1, PAK1, and OCT4 mRNA" should be "increased levels of MYL9, ITGB1, and PAK1 mRNA"

1st Revision - authors' response

20th Mar 19

Referee#1:

In this study the authors identify RASSF1A, the Hippo kinase regulator, as a regulator of actin monomer levels in the nuclear compartment via interacting with exportin 6, the specific actin export factor, as well as with the Ran GTPase. This leads to changes in actin-dependent gene regulation and may have functional effects on cell adhesion, all supposedly mediated by subsequent changes in the activity of MRTF-A and SRF. Finally, the authors try to correlate their findings with clinical data of invasive carcinoma. Overall, the findings are interesting since they identify a novel player in the nuclear actin-MRTF module, however, several further controls are needed.

We would like to thank the reviewer for recognising the novelty of our work. Below there is a detailed description of all the experiments and changes performed to answer their concerns.

1. Fig.2C the quantification of effective-knockdowns should be included; RASSF1A decrease resulted from MST2 needs to be confirmed via Western blot.

We now include these data on Figs EV1A and EV2C. The lack of RASSF1A immunofluorescence is due to redistribution away from the NE (Fig 2C), as the total and nuclear protein levels of RASSF1A do not change in MST2-depleted cells (EV2C and EV2D).

2. It is claimed that SARAH domain is responsible for the interaction, therefore, in Fig. 3B it is necessary to rule out the involvement of RA domain by introducing a 288-340 mutation.

We agree with the reviewer that the truncated RASSF1A expressing only the SARAH domain is missing, however the SARAH domain alone is not stable and expression was ineffective (as previously found). In light of this we have reworded the text to fit the data by describing the observation at the SARAH domain being 'required' for the interaction rather than responsible.

3. Fig. 3C as the co-immunoprecipitation assay showed, Ran recruitment to the RASSF1A/XPO6/RanGTP complex is totally abolished. Could this be seen in staining as well? The same concern also exists in the Fig. 3D, disturbed formation of the complex after MST2 depletion. Additionally, since MST2 is not involved in the RASSF1A/XPO6/RanGTP complex, how come it affects the complex formation? What is the mechanism?

This is an important aspect of the manuscript that we did not originally focus on and also a concern of Rev 2 (point 2) and Rev 3 (point 1), however we now provide further experiments to elucidate a potential mechanism where we expand on XPO6 IP +/- RASSF1A and RASSF1A IP +/- MST2 (new Fig 3C,E). We provide additional XPO6 IP (+/- RASSF1A and +/- MST2) together with MST2 IP (+/- RASSF1A and +/- XPO6) to delineate hierarchy of associations and put the complexes in context of the NE association. These additional data indicate that XPO6 and RAN interact independently of MST2, with limited involvement of RASSF1A (or at a reduced level). However, the dependence of XPO6/RAN stability on RASSF1A appears to be restricted to MST2 being present, which we interpret to be due to NE recruitment. We also further support RASSF1A involvement by employing a recombinant RAN binding assay where XPO6 association is reduced in the absence of RASSF1A (new EV3B). We have modified the text to explain this mechanism as below;

....We further investigated the role of RASSF1A on the association of RAN with XPO6. Most strikingly, RASSF1A appeared to be required to support the XPO6/RAN complex, as siRNA-mediated knockdown of RASSF1A decreased association between XPO6 and RAN (Fig 3C). Expression of RASSF1A in MDA-MB-231 cells significantly enhances the association of XPO6 with RAN (EV3A). We validated this requirement for RASSF1A with a GST pull-down assay using recombinant GST-RAN and lysates from siCTRL or siRASSF1A-transfected HeLa cells (EV3B). Notably, XPO6 co-immunoprecipitation (IP) indicated that the XPO6/RAN complex with RASSF1A also includes MST2, suggesting potential recruitment to the NE via the RASSF1A-MST2 interaction (Fig 3C). Depletion of MST2 expression using siRNA did not affect XPO6/RAN as dramatically as siRASSF1A, but did reduce XPO6/RASSF1A, which we believe implies that RASSF1A may be required for stabilising XPO6/RAN at the NE, i.e. in an MST2 dependent manner, but the nucleoplasmic XPO6/RAN pool may be less dependent on RASSF1A (Fig 3C). This is supported by the fact that the RASSF1A interaction with XPO6/RAN is also dependent on MST2, and therefore NE localisation (Fig 3D). To verify this mechanism, we explored MST2 associated proteins by IP and found that XPO6/RAN interaction with MST2 was RASSF1A dependent whereas the RASSF1A/RAN interaction with MST2 did not require XPO6 (Fig 3E), confirming our hypothesis that XPO6/RAN complex is stabilised by MST2/RASSF1A interaction.

Taken together, our results show interaction of XPO6 with RAN can occur independently of RASSF1A, but a pool of XPO6/RAN is stabilised by RASSF1A in a MST2 dependent manner at the NE.

We have also looked at whether this model could be supported by IF (Fig for reviewers 1, below). Suppression of RASSF1A expression does appear to reduce the nuclear staining of RAN, and as total levels of RAN are unaffected (Fig 3C,E), this implies redistribution to the cytoplasm. As staining of XPO6 is unaffected this supports the biochemical data that the XPO6/RAN complex is reduced. Moreover, it also suggests that RASSF1A might be important to maintain RAN.GTP levels, and RAN.GDP is cytoplasmic, and this may impact on the complex formation. As this

requires much further thorough investigation we feel it is too preliminary to include in this story, but have included the following statement in the discussion.

.. the reduced involvement of RASSF1A in the XPO6/RAN complex in the absence of MST2 suggests that RAN/XPO6 exists independently of RASSF1A in the nucleoplasm. This means that XPO6/RAN complexes may be contextually distinct from RASSF1A/RAN/XPO6 and could involve differences in substrate loading, RAN GDP/GTP loading or post translational modifications of RAN [Guttler and Gorlich, 2011 EMBOJ][Dallol et al, 2007] [de Boor et al. 2015 PNAS][Bompard et al 2010 jcb].

Figure for Reviewers 1

4. Fig.4 is not entirely convincing. All conclusions on nuclear actin levels are based on biochemical fractionations, which can be often problematic. Besides, effects of siRASSF1A need to be rescued by siRNA resistant wt versus SARAH domain deletion derivatives! This is important and necessary. The biochemical analysis of cleat actin levels needs to be complemented by nuclear actin visualizations either with endogenous actin or at least using GFP-actin.

We originally performed fractionations as the visualization of endogenous nuclear actin is particularly challenging and usage of phalloidin could stabilise actin filaments (Coluccio and Tilney, 1984; Visegrády et al. 2004). It has been shown also that phalloidin might not bind to all the actin structures present in the nucleus as it requires at least seven actin subunits for binding to F-actin, neglecting labelling of short F-actin polymers (Kristó et al., 2016). Moreover, overexpression of epitope-tagged actin can be problematic, as even a subtle change in the amount of actin can interfere with the physiological actin dynamics, and could trigger actin polymerization by itself (Mounier et al., 1997; Ballestrem et al., 1998). However, we did try to visualise nuclear actin filaments (Phalloidin) and actin monomers (DNase I) and although we did not observe any change of the levels of filamentous actin in the nucleus, we now show increased monomeric actin in RASSF1A-depleted cells. These data are now shown in Fig 4C and EV4F.

It is indeed necessary to validate our results with rescue experiments. Therefore as suggested, we add a new set of experiments in Fig EV4A-C where the effects of RASSF1A silencing on nuclear actin and profilin levels were rescued by co-transfection of siRASSF1A with either SARAH-containing RASSF1A derivatives (but not by SARAH-domain truncated) or plasmids encoding for

XPO6. Increase of nuclear actin and profilin in RASSF1A silenced cells was also rescued with a siRNA resistant RASSF1A expression plasmid.

5. Could the effect of RASSF1A depletion on nuclear actin levels be rescued by overexpression of XPO6? This would more directly confirm XPO6 is downstream of RASSF1A. In fact, rescue and reconstitution studies would be highly desirable.

This is a very interesting point and we thank the reviewer for this suggestion. We now include the results in Fig. EV4B

6. How does RASSF1A affect MRTF-A? The authors should study MRTF-A (also known as MAL, or MKL1!) actin interactions directly in the presence or absence of RASSF1A.

We found that increased nuclear G-actin resulted from RASSF1A silencing leads to increased actin-MRTF-A interactions. We include these data in Fig. EV5C.

7. What happens to nuclear F-actin in RASSF1A silenced cells? Did the author make efforts to visualize endogenous nuclear F-actin? This would be interesting and provide more and better mechanistic insight, since it has been shown in several studies that nuclear actin assembly regulates MRTF-A or MKL1 or MAL activity. It is also well established by now that nuclear actin is dynamically polymerized by various cellular cues and inputs. Hence, does an increase in nuclear actin levels lead to actin polymerization or changes of nuclear F-actin in this compartment?

This is very interesting point and related to point 4. Probing endogenous F-actin by using fluorescently labelled phalloidin showed no difference of nuclear actin filaments between control RNA-treated cells and cells silenced for RASSF1A, as shown in Fig EV4F. Interestingly, visualisation of G-actin using fluorescently labelled DNase I revealed increase nuclear G-actin in RASSF1A silenced cells (Fig 4C). This is in agreement with our data showing cytoplasmic MRTF-A (Fig 5C) as increased nuclear G-actin promotes the nuclear export of MRTF-A (Vartiainen et al. 2007).

Referee#2:

The manuscript by Chatzifrangkeskou et al. describes a novel role for RASSF1A tumor suppressor gene in regulating nuclear export of actin, and thus also MRTF-A-SRF mediated transcription. RASSF1A is a member of a protein family acting upstream of the Hippo-pathway, binding directly to the MST1 and 2 kinases of the Hippo pathway, and its inactivation is implicated in the development of various cancers.

In the present manuscript, the authors broaden the functional aspects of RASSF1A by linking it to regulation of nuclear actin levels. Actin has been shown to shuttle in and out of the nucleus with the aid of transport factor Importin-9 and Exportin-6, but further regulators are poorly characterized. The authors first show that a pool of RASSF1 can be found on the nuclear envelope, and that this localization is dependent on MST2. Based on the previous data on the interaction between RASSF1 and the Ran-GTPase, which is the major driver of energy dependent nuclear transport, the authors then identify an interaction between RASSF1 and Exportin-6, and surprisingly find that RASSF1 is required for the interaction between Exportin-6 and Ran. In cells, depletion of RASSF1 leads to increased nuclear accumulation of actin and profilin, as well as to decreased expression of specific genes previously linked to nuclear actin, cell adhesion defects and to decreased nuclear localization of the transcription coactivator MRTF-A and consequently decreased SRF activity. Finally, the expression of SRF and RASSF1A are correlated in various cancer cells.

The outset for the manuscript is very interesting, and the idea that the nuclear actin pool, and consequently gene expression, would be regulated by a tumour suppressor is exciting. However, the mechanism by which RASSF1A operates here remains somewhat unclear, and further experiments as well as quantifications would be needed to support the key hypothesis.

Major concern

1. First, how is the NE localization of RASSF1 related to whole business? How would the NE localized RASSF1A contribute to the formation of the Exportin-6-Ran-actin export complex, especially since MST2 does not seem to be in the complex? MST2 seems to be required for RASSF1A localization to the NE, but can it be excluded that it also regulates the nuclear localization of RASSF1A? In these experiments, the subcellular localization of RASSF1A should be analyzed quantitatively from a large number of cells (or by fractionation) to draw any general conclusions. Analyzing actin distribution in MST2 depleted cells (in case it does not affect nuclear localization of RASSF1A per se) would tell, if the NE localization of RASSF1A is important for regulating nuclear actin.

We thank the reviewer for this suggestion. The nuclear localization of RASSF1A is not affected by the depletion of MST2 as evaluated by both immunofluorescent staining and fractionation (Fig EV2D, EV4D). However, the MST2-dependent RASSF1A NE localisation is important for the regulation of nuclear actin levels as shown by the altered nuclear actin levels in MST2-depleted cells (Fig EV4D).

2. Second, I am very surprised by the huge effect that RASSF1 depletion has on Exportin-6-Ran interactions (figure 3C), especially since it seems that the depletion of RASSF1 is not very efficient. Moreover, Exportin-6 has been shown to interact directly, like most other nuclear transport factors, with Ran (shown for example in Stuken et al. 2003). Why would the interaction in cells then require an additional factor? The IPs in figures 3C and D should be carefully quantified from several experiments to draw firm conclusions on these experiments. Related to this point, did the authors check the subcellular distribution of Exportin-6 in RASSF1A depleted cells?

This is essentially the same issue raised by Rev1 (point 3) and Rev 3 (point 1), and an important aspect of the manuscript that we did not originally focus on, however we now provide further experiments to elucidate a potential mechanism where we expand on XPO6 IP +/- RASSF1A and RASSF1A IP +/- MST2 (new Fig 3C,E). We provide additional XPO6 IP (+/- RASSF1A and +/- MST2) together with MST2 IP (+/- RASSF1A and +/- XPO6) to delineate hierarchy of associations and put the complexes in context of the NE association. These additional data indicate that XPO6 and RAN interact independently of MST2, with limited involvement of RASSF1A (or at a reduced level). However, the dependence of XPO6/RAN stability on RASSF1A appears to be restricted to MST2 being present, which we interpret to be due to NE recruitment. We also further support RASSF1A involvement by employing a recombinant RAN binding assay where XPO6 association is reduced in the absence of RASSF1A (new EV3B). We have modified the text to explain this mechanism as below;

....We further investigated the role of RASSF1A on the association of RAN with XPO6. Most strikingly, RASSF1A appeared to be required to support the XPO6/RAN complex, as siRNA-mediated knockdown of RASSF1A decreased association between XPO6 and RAN (Fig 3C). Expression of RASSF1A in MDA-MB-231 cells significantly enhances the association of XPO6 with RAN (EV3A). We validated this requirement for RASSF1A with a GST pull-down assay using recombinant GST-RAN and lysates from siCTRL or siRASSF1A-transfected HeLa cells (EV3B). Notably, XPO6 co-immunoprecipitation (IP) indicated that the XPO6/RAN complex with RASSF1A also includes MST2, suggesting potential recruitment to the NE via the RASSF1A-MST2 interaction (Fig 3C). Depletion of MST2 expression using siRNA did not affect XPO6/RAN as dramatically as siRASSF1A, but did reduce XPO6/RASSF1A, which we believe implies that RASSF1A may be required for stabilising XPO6/RAN at the NE, i.e. in an MST2 dependent manner, but the nucleoplasmic XPO6/RAN pool may be less dependent on RASSF1A (Fig 3C). This is supported by the fact that the RASSF1A interaction with XPO6/RAN is also dependent on MST2, and therefore NE localisation (Fig 3D). To verify this mechanism, we explored MST2 associated proteins by IP and found that XPO6/RAN interaction with MST2 was RASSF1A dependent whereas the RASSF1A/RAN interaction with MST2 did not require XPO6 (Fig 3E), confirming our hypothesis that XPO6/RAN complex is stabilised by MST2/RASSF1A interaction.

Taken together, our results show interaction of XPO6 with RAN can occur independently of RASSF1A, but a pool of XPO6/RAN is stabilised by RASSF1A in a MST2 dependent manner at the NE.

As suggested, we have also looked at XPO6 and RAN distribution (Fig for reviewers 1, above). Suppression of RASSF1A expression does appear to reduce the nuclear staining of RAN, and as total levels of RAN are unaffected (Fig 3C,E), this implies redistribution to the cytoplasm. As staining of XPO6 is unaffected this supports the biochemical data that the XPO6/RAN complex is reduced. Moreover, it also suggests that RASSF1A might be important to maintain RAN.GTP levels, and RAN.GDP is cytoplasmic, and this may impact on the complex formation. As this requires much further thorough investigation we feel it is too preliminary to include in this story, but have included the following statement in the discussion.

.. the reduced involvement of RASSF1A in the XPO6/RAN complex in the absence of MST2 suggests that RAN/XPO6 exists independently of RASSF1A in the nucleoplasm. This means that XPO6/RAN complexes may be contextually distinct from RASSF1A/RAN/XPO6 and could involve differences in substrate loading, RAN GDP/GTP loading or post translational modifications of RAN [Guttler and Gorlich, 2011 EMBOJ][Dallol et al, 2007] [de Boer et al. 2015 PNAS][Bompard et al 2010 jcb].

3. Third, the results in figures 4C and 4D on MDA-MB-231 cells are very clear, and the data seems to imply that due to the lack of RASSF1A, these cells have relatively high levels of nuclear actin. However, the Treisman-lab has shown that in these cells, MRTF-A is actually nuclear (Medjkane et al. 2009, NCB), which is exactly the opposite that you would expect, and not in agreement with the subsequent results in figure 5. What is the explanation for this?

Indeed, the report from Medjkane et al. showed that MDA-MB-231 cells express high levels of RhoA-GTP and therefore MRTF-A is predominantly nuclear. However, RhoA promotes the polymerization of G to F-actin in the cytoplasm as well as in the nucleus. We speculate that the high nuclear actin levels in MDA-MB-231 exist in filamentous state due to the active RhoA signalling. Thus, depletion of G-actin impairs the MRTF-A nuclear export and promotes its nuclear import (Vartiainen et al. 2007; Pawlowski et al. 2010). To address this, we did inhibit Rho/ROCK signalling using Y27632, and as can be seen in figure for reviewers 2, although nuclear actin in MDA-MB-231 is high and MRTF-A nuclear inhibition of ROCK (which should reduce filamentous F-actin) increases G-actin and is concomitant with MRTF-A export.

Figure for Reviewers 2: Confocal images of G-actin (DNase I) and MRTF-A in control and ROCK inhibitor Y27632-treated-MDA-MB-231 cells.

4. Fourth, it is surprising how well the Importin-9 silencing can "rescue" the effects of RASSF1A silencing on gene expression, adhesion and SRF activation in figure 5. However, key controls are missing. The authors should first of all convincingly demonstrate that Importin-9 is appropriately silenced, since the blot in S3A (of nuclear fractions only) is not very clear. They should then demonstrate that importin-9 silencing can actually "rescue" the nuclear actin levels back to normal with fractionation.

As suggested we now validated the silencing of IPO9 by both qPCR and immunoblotting (EV5A-B).

In figure 1D, the PLA experiment does add too much information, and if kept in the manuscript, should be quantified.

We agree with the reviewer and we now exclude these data from the manuscript.

Figure 1E would have benefitted from a positive control for the digitonin treatment (e.g. some outer nuclear membrane protein) to prove that plasma membrane is permeabilized, since the RASSF1A staining in the cytoplasm does not look very convincing.

We now add the Fig EV1F, in which we used α -Tubulin as a control of plasma membrane permeabilisation.

Minor points

In quantification of figure 4C, it is not clear what the data is relative to.

At several places, the authors talk about "expression", when they should be talking about localization. In figure 1A, the white dashed line is in the wrong place.

The materials and methods contains stuff that is not in the manuscript, e.g. on fluorouridine.

On page the sentence "Lamin A/C and B1 both contribute to NE integrity, with Lamin B constituting more stable filaments and Lamin A/C more responsible for stable rigidity." is wrong and does not contain references.

On page 7, depletion of IPO9 inhibits nuclear import of actin, NOT promotes export.

On page 7, last sentence on first paragraph does not reflect the results, since not all genes were

increased.

In discussion, the first reference is not optimal, since for example the nuclear import pathway for actin was not discovered, when that paper was published.

We have corrected the typos and inconsistencies.

Referee #3:

In this manuscript, the authors demonstrated that RASSF1A (Ras association domain family 1 isoform A), a tumour suppressor, functions as a novel regulator of actin nucleocytoplasmic trafficking to regulate nuclear actin levels. They found that RASSF1A localizes to nuclear envelope, and supports the binding between exportin-6 (XPO6) and RAN GTPase to modulate the nuclear actin export. Furthermore, they showed that RASSF1A is involved in the regulation of MRTF-A, a coactivator of the SRF transcription factor. This is an interesting paper containing potentially important findings and general significance. However, the current manuscript does not provide the mechanistic insight into how RASSF1A participates in nucleocytoplasmic transport process, especially, the formation of XPO6/RAN complex.

Specific comments:

(1) It has been already demonstrated that XPO6 directly binds to RAN (Q69L), using recombinant proteins (Stuven et al, 2003). So, how does RASSF1A affect the formation of XPO6/RAN complex? Since this is one of the most striking findings in this study, the authors need to clarify the role of RASSF1A.

This was similar to the comment by Rev1 (point 3) and Rev2 (point 2) and was important to clarify further as an aspect of the manuscript that we did not originally focus on, however we now provide further experiments to elucidate a potential mechanism where we expand on XPO6 IP +/- RASSF1A and RASSF1A IP +/- MST2 (new Fig 3C,E). We provide additional XPO6 IP (+/- RASSF1A and +/- MST2) together with MST2 IP (+/- RASSF1A and +/- XPO6) to delineate hierarchy of associations and put the complexes in context of the NE association. These additional data indicate that XPO6 and RAN interact independently of MST2, with limited involvement of RASSF1A (or at a reduced level). However, the dependence of XPO6/RAN stability on RASSF1A appears to be restricted to MST2 being present, which we interpret to be due to NE recruitment. We also further support RASSF1A involvement by employing a recombinant RAN binding assay where XPO6 association is reduced in the absence of RASSF1A (new EV3B). We have modified the text to explain this mechanism as below;

....We further investigated the role of RASSF1A on the association of RAN with XPO6. Most strikingly, RASSF1A appeared to be required to support the XPO6/RAN complex, as siRNA-mediated knockdown of RASSF1A decreased association between XPO6 and RAN (Fig 3C). Expression of RASSF1A in MDA-MB-231 cells significantly enhances the association of XPO6 with RAN (EV3A). We validated this requirement for RASSF1A with a GST pull-down assay using recombinant GST-RAN and lysates from siCTRL or siRASSF1A-transfected HeLa cells (EV3B). Notably, XPO6 co-immunoprecipitation (IP) indicated that the XPO6/RAN complex with RASSF1A also includes MST2, suggesting potential recruitment to the NE via the RASSF1A-MST2 interaction (Fig 3C). Depletion of MST2 expression using siRNA did not affect XPO6/RAN as dramatically as siRASSF1A, but did reduce XPO6/RASSF1A, which we believe implies that RASSF1A may be required for stabilising XPO6/RAN at the NE, i.e. in an MST2 dependent manner, but the nucleoplasmic XPO6/RAN pool may be less dependent on RASSF1A (Fig 3C). This is supported by the fact that the RASSF1A interaction with XPO6/RAN is also dependent on MST2, and therefore NE localisation (Fig 3D). To verify this mechanism, we explored MST2 associated proteins by IP and found that XPO6/RAN interaction with MST2 was RASSF1A dependent whereas the RASSF1A/RAN interaction with MST2 did not require XPO6 (Fig 3E), confirming our hypothesis that XPO6/RAN complex is stabilised by MST2/RASSF1A interaction.

Taken together, our results show interaction of XPO6 with RAN can occur independently of RASSF1A, but a pool of XPO6/RAN is stabilised by RASSF1A in a MST2 dependent manner at the NE.

As suggested, we have also looked at XPO6 and RAN distribution (Fig for reviewers 1, above). Suppression of RASSF1A expression does appear to reduce the nuclear staining of RAN, and as total levels of RAN are unaffected (Fig 3C,E), this implies redistribution to the cytoplasm. As staining of XPO6 is unaffected this supports the biochemical data that the XPO6/RAN complex is reduced. Moreover, it also suggests that RASSF1A might be important to maintain RAN.GTP levels, and RAN.GDP is cytoplasmic, and this may impact on the complex formation. As this

requires much further thorough investigation we feel it is too preliminary to include in this story, but have included the following statement in the discussion.

.. the reduced involvement of RASSF1A in the XPO6/RAN complex in the absence of MST2 suggests that RAN/XPO6 exists independently of RASSF1A in the nucleoplasm. This means that XPO6/RAN complexes may be contextually distinct from RASSF1A/RAN/XPO6 and could involve differences in substrate loading, RAN GDP/GTP loading or post translational modifications of RAN [Guttler and Gorlich, 2011 EMBOJ] [(Dallol et al, 2007) [de Boor et al. 2015 PNAS] [Bompard et al 2010 jcb].

The authors could perform a binding assay using recombinant proteins to examine how the presence of RASSF1A affects the binding between XPO6 and RAN. In addition, the authors should examine the binding between XPO6 and RAN in MDA-MB-213 cells by co-immunoprecipitation (as in Figure 3C). It may also be possible that the knockdown of RASSF1A causes the subcellular translocation of XPO6 or RAN.

We thank reviewer for this suggestion and we include this data on Fig EV3B where we performed GST-RAN pull down assay using lysates from siCTRL and siRASSF1A-treated cells. We also showed that the binding of XPO6 to RAN is significantly enhanced upon expression of RASSF1A in MDA-MB-231 cells as shown in Fig EV3A. The knockdown of RASSF1A does not alter the subcellular localisation of XPO6 and RAN as showed on page 1 (Rev1, point 3).

(2) Figure 3C: The band intensity of XPO6 in IP sample seems too weak. Also, in Figure 3D, the band intensity of RASSF1A is weak as well. Therefore, it is not convincing if these antibodies could precipitate their antigens.

Western blots have been replaced with new IPs from new experiments to clarify.

(3) Figure 4A: The effect of siRASSF1A on actin/profilin nuclear export appears to be relatively modest, considering that XPO6-RAN interaction is almost completely disrupted by the same treatment (Figure 3C). This could be due to the higher concentration of actin/profilin in cytoplasm, as the authors stated. However, to clarify this, authors should examine the effect of XPO6 knockdown on the actin/profilin nuclear export.

Apologies if we misunderstand but we interpret this critique to be related to the extend to which nuclear actin and profilin levels are mainlined upon siRASSF1A. We do feel this are significant and are quantified in associated bars graphs to reflect this. In figure for reviewers 3, we provide additional data on nuclear/cytoplasmic levels of actin and profilin in the absence of XPO6 to demonstrate that the effect of siRASSF1A is similar to that achieved by siXPO6 alone, but as this is not novel we are not including this in the manuscript. We hope this addresses the concerns.

Figure for Reviewers 3

(4) Page 7:

"increased levels of MYL9, ITGB1, PAK1, and OCT4 mRNA" should be "increased levels of MYL9, ITGB1, and PAK1 mRNA"

This is now corrected.

Thank you for submitting a revised version of your manuscript. It has now been seen by all original referees, which are included below for your information.

As you will see from the comments, while reviewers #1 and #3 find that their concerns have been sufficiently addressed, referee #2 notes that key conclusions of the manuscript have changed in the revision and requests a thorough quantification of the data to convincingly support the proposed mechanism. I agree with reviewer #2 that it is crucial to add this information to the manuscript before it can be accepted for publication here. Please also provide an explanation for this change in the conclusions in your point-by-point letter. Additionally, there are several issues with the submitted source data:

1. Fig 1C source data appear to be mislabeled.
2. Fig 3B IB Myc - image overcontrasted or bad resolution. Are the XPO6 input bands from the same blot as the IP, as they do not match the source data?
3. Dimensions of panels appear to be altered in Fig 3A, 3C, 3D, 4A, EV1C, EV2A (lamin A/C), EV2C, EV3B (RASSF1A), EV4C, EV4D (lamin A/C), EV4E (GAPDH), EV4G, EV5B
4. In Fig 4D, profilin panel does not fit to the source data
5. In Fig EV1A, RASSF1A panel does not fit to the source data
6. Several source data panels are mislabeled (EV1E should be EV1F, EV2B should be EV2A, EV2A should be EV2C, EV4D should be EV4E and vice versa, EV5D should be EV5C)
7. Background has been removed in Fig EV4A Myc panel
8. Contrast has been changed in an unusual manner for Fig EV4B (actin and XPO6 panels), EV4G (GAPDH and RASSF1A panel), EV5B (IPO9 panel). In general, contrast has been adjusted in many blots for unclear reasons.

There are also additional editorial issues that I would like to ask you to address in the revised manuscript:

REFEREE REPORTS:

Referee #1:

The authors have done a very good job in revising their manuscript. I cartoon illustrating the role of RASSF1A in nuclear-cytoplasmic shuttling of actin and MRTF would help to guide the reader possibly.

Referee #2:

In their revised Chatzifrangkeskou et al. have extensively modified their manuscript according to the criticism by the reviewers. Indeed, these revisions have made parts of the manuscript stronger. However, most of my main concerns still remain.

To elucidate the mechanism by which RASSF1 (and MST2) regulate nuclear actin levels, the authors have done further experiments, and e.g. more co-immunoprecipitation experiments. In the first submission, RASSF1 depletion (which was only partial) did completely abolish the interaction between Exportin-6 and Ran, and MST2 was not detected in the complex. Now in the revised manuscript, the data shows decreased interaction between Exportin-6 and Ran in RASSF1 depleted cells, and also MST2 can be detected in the complex. Because of these discrepancies, it is **ABSOLUTELY CRUCIAL** that the authors quantify the results from at least three independent experiments and also show the original data as uncropped and unprocessed blots (see my comment also below).

There is a general lack of quantification throughout the manuscript. For instance, the authors have now examined actin distribution upon MST2 depletion, but the data is not quantified. Similarly, the authors have examined nuclear actin levels upon Importin-9 depletion, but the data is not quantified. Moreover, although the qPCR analysis seems to suggest that Importin-9 depletion worked, it is impossible to judge this at the protein level, and this is general trend in the manuscript. I suggest that the authors show and quantify their depletion efficiencies for each case.

Finally, the blots shown in uncropped/unprocessed blots do not appear to be either, since it is clear

that the blots have been cropped, and also somehow adjusted, since for example the background seems very similar in most cases (even in blots that should have different exposure times). There are also some inconsistencies in labeling the blots, so I suggest that you check these carefully.

Referee #3:

The authors have addressed most of my concerns in a satisfactory manner.

2nd Revision - authors' response

23rd Apr 2019

Please see next page.

EDITOR:

As you will see from the comments, while reviewers #1 and #3 find that their concerns have been sufficiently addressed, referee #2 notes that key conclusions of the manuscript have changed in the revision and requests a thorough quantification of the data to convincingly support the proposed mechanism. I agree with reviewer #2 that it is crucial to add this information to the manuscript before it can be accepted for publication here. Please also provide an explanation for this change in the conclusions in your point-by-point letter.

RESPONSE:

Further thorough quantification is now provided. Importantly the conclusions have not changed from the original submission in which we stated “Taken together, we have identified a previously unknown mechanism by which the nuclear actin pool is regulated and uncovered a previously unknown link of RASSF1A and MTRF/SRF in tumour suppression.” The reviewer asked us to provide further data on the mechanism by exploring the regulatory complex in more detail. In this regard, we found that MST2 can be at low levels in the complex - this however was not the main point of the article, was not presented as a conclusion in the summary or mentioned in the title. We discussed the finding of MST2 in the complex in detail in the original rebuttal below;

Reviewer 2 point2. Second, I am very surprised by the huge effect that RASSF1 depletion has on Exportin-6-Ran interactions (figure 3C), especially since it seems that the depletion of RASSF1 is not very efficient. Moreover, Exportin-6 has been shown to interact directly, like most other nuclear transport factors, with Ran (shown for example in Stuvén et al. 2003). Why would the interaction in cells then require an additional factor? The IPs in figures 3C and D should be carefully quantified from several experiments to draw firm conclusions on these experiments. Related to this point, did the authors check the subcellular distribution of Exportin-6 in RASSF1A depleted cells?

This is essentially the same issue raised by Rev1 (point 4) and Rev 3 (point 1), and an important aspect of the manuscript that we did not originally focus on, however we now provide further experiments to elucidate a potential mechanism where we expand on XPO6 IP +/- RASSF1A and RASSF1A IP +/- MST2 (new Fig 3C,E). We provide additional XPO6 IP (+/- RASSF1A and +/- MST2) together with MST2 IP (+/- RASSF1A and +/- XPO6) to delineate hierarchy of associations and put the complexes in context of the NE association. These additional data indicate that XPO6 and RAN interact independently of MST2, with limited involvement of RASSF1A (or at a reduced level). However, the dependence of XPO6/RAN stability on RASSF1A appears to be restricted to MST2 being present, which we interpret to be due to NE recruitment. We also further support RASSF1A involvement by employing a recombinant RAN binding assay where XPO6 association is reduced in the absence of RASSF1A (new EV3B). We have modified the text to explain this mechanism as below;

Referee #2:

In their revised Chatzifrangkeskou et al. have extensively modified their manuscript according to the criticism by the reviewers. Indeed, these revisions have made parts of the manuscript stronger. However, most of my main concerns still remain.

RESPONSE:

We thank the reviewer for their recognition that the additional data we present addressed their concerns and made the manuscript stronger. In an effort to satisfy their remaining concerns we have addressed these explicitly below.

Point 1. To elucidate the mechanism by which RASSF1 (and MST2) regulate nuclear actin levels, the

authors have done further experiments, and e.g. more co-immunoprecipitation experiments. In the first submission, RASSF1 depletion (which was only partial) did completely abolish the interaction between Exportin-6 and Ran, and MST2 was not detected in the complex. Now in the revised manuscript, the data shows decreased interaction between Exportin-6 and Ran in RASSF1 depleted cells, and also MST2 can be detected in the complex.

As described in the original rebuttal, we further investigated the complex as a specific request to elucidate the mechanism – our overall conclusions have not changed. The original and the revised manuscript described the same – decreased interaction between XPO6 and RAN in RASSF1 depleted cells, this has not changed in the resubmission as the reviewer appears to suggest. We do now identify a low level of MST2 in the complex which was undetectable by standard approaches used previously and required super sensitive ECL methods, which we only applied to address the mechanistic question raised by the reviewer. The presence of MST2 in the complex clarifies the original conclusion but again does not change the novel process of nuclear actin regulation we describe.

Because of these discrepancies, it is ABSOLUTELY CRUCIAL that the authors quantify the results from at least three independent experiments and also show the original data as uncropped and unprocessed blots (see my comment also below).

We apologise for not making this clear in the figure legends and not making the PCR and statistical statement in the methods more comprehensive. All experiments were in keeping with standard scientific rigor and representative of at least n = 3 experiments. For figure 3C and 3D we now provide additional quantification of the representative IP western blots from 3 independent experiments. In the R1 version EV3C, EV3D we had demonstrated the original uncropped and unprocessed blot of Fig3C (scanned at 16bit grayscale), Fig 3C blots are reproduced below for editorial consideration have been scanned again at 48bit colour to demonstrate the scanned images have not been manipulated.

Point 2. There is a general lack of quantification throughout the manuscript. For instance, the authors have now examined actin distribution upon MST2 depletion, but the data is not quantified. Similarly, the authors have examined nuclear actin levels upon Importin-9 depletion, but the data is not quantified. Moreover, although the qPCR analysis seems to suggest that Importin-9 depletion worked, it is impossible to judge this at the protein level, and this is general trend in the manuscript. I suggest that the authors show and quantify their depletion efficiencies for each case.

This is an unfair comment as we have provided extensive quantification for the main data of Fig 4, 5 and 6 in both the original submission and the revised version. We also provided quantification for knockdown experiments in EV1A, EV2C and EV5A. We are happy to provide the further requested quantification of actin levels with siMST2 (new Fig EV4D) and siIPO9 (new Fig EV5B) to support the clearly apparent changes in protein levels. The data in Fig EV5B demonstrated an evident decrease in IPO9 protein level (red boxes below), which we felt clearly reflected the changes in rtPCR levels of IPO9, is now further supported by quantification of this control (from n=3 experiments) as requested in new fig EV5B.

Point 3. Finally, the blots shown in uncropped/unprocessed blots do not appear to be either, since it is clear that the blots have been cropped, and also somehow adjusted, since for example the background seems very similar in most cases (even in blots that should have different exposure times).

Apologies, but is not clear exactly what is required here. Does the reviewer want to see the original films that were exposed? If so an example of this is below. Blots are a true representation of the original scanned imaged and have not been manipulated. The background emanates from the film that the ECL treated blots were exposed to, therefore should be the same in all cases. Any cropping was to reduce overall file size but maintained the representation of the original gel-transferred filter relevant rather than the entire film – and is appropriate for journals.

There are also some inconsistencies in labeling the blots, so I suggest that you check these carefully.

We thank the reviewer for pointing out errors and these have now been corrected.

EDITOR:

Additionally, there are several issues with the submitted source data:

- 1. Fig 1C source data appear to be mislabeled.*
- 2. Fig 3B IB Myc - image overcontrasted or bad resolution. Are the XPO6 input bands from the same blot as the IP, as they do not match the source data?*
- 3. Dimensions of panels appear to be altered in Fig 3A, 3C, 3D, 4A, EV1C, EV2A (lamin A/C), EV2C, EV3B (RASSF1A), EV4C, EV4D (lamin A/C), EV4E (GAPDH), EV4G, EV5B*
- 4. In Fig 4D, profilin panel does not fit to the source data*
- 5. In Fig EV1A, RASSF1A panel does not fit to the source data*
- 6. Several source data panels are mislabeled (EV1E should be EV1F, EV2B should be EV2A, EV2A should be EV2C, EV4D should be EV4E and vice versa, EV5D should be EV5C)*
- 7. Background has been removed in Fig EV4A Myc panel*
- 8. Contrast has been changed in an unusual manner for Fig EV4B (actin and XPO6 panels), EV4G (GAPDH and RASSF1A panel), EV5B (IPO9 panel). In general, contrast has been adjusted in many blots for unclear reasons.*

RESPONSE:

We thank the Editor for pointing out errors and these have now been corrected.

Thanks very much for incorporating the final changes into the manuscript. I am now happy to inform you that your manuscript has been accepted for publication in The EMBO Journal. Congratulations on a nice study!

Corresponding Author Name: Eric O'Neill
 Journal Submitted to: EMBO Journal
 Manuscript Number: EMBOJ-2018-101168R